# ZERO-SHOT TASK ADAPTATION BY HOMOICONIC META-MAPPING

## ABSTRACT

How can deep learning systems flexibly reuse their knowledge? Toward this goal, we propose a new class of challenges, and a class of architectures that can solve them. The challenges are meta-mappings, which involve systematically transforming task behaviors to adapt to new tasks zero-shot. The key to achieving these challenges is representing the task being performed in such a way that this task representation is itself transformable. We therefore draw inspiration from functional programming and recent work in meta-learning to propose a class of Homoiconic Meta-Mapping (HoMM) approaches that represent data points and tasks in a shared latent space, and learn to infer transformations of that space. HoMM approaches can be applied to any type of machine learning task, including supervised learning and reinforcement learning. We demonstrate the utility of this perspective by exhibiting zero-shot remapping of behavior to adapt to new tasks.

## 1 INTRODUCTION

Humans are able to use and reuse knowledge more flexibly than most deep learning models can (Lake et al., 2017; Marcus, 2018). The problem of rapid learning has been partially addressed by meta-learning systems (Santoro et al., 2016; Finn et al., 2017; 2018; Stadie et al., 2018; Botvinick et al., 2019, see also Section 7). However, humans can use our knowledge of a task to flexibly adapt when the task changes. In particular, we can often perform an altered task zero-shot, that is, without seeing any data at all. For example, once we learn to play a game, we can immediately switch to playing in order to lose, and can perform reasonably on our first attempt.

One fundamental reason for this is that humans are aware of what we are trying to compute and why. This allows us to adapt our task representations to perform an altered task. By contrast, most prior work on zero-shot learning relies on generating a completely new representation for a new task, e.g. from a natural language description (Socher et al., 2013; Oh et al., 2017, e.g.). These systems must therefore throw away much of their prior knowledge of the task and regenerate it from scratch, rather than just transforming what needs to be transformed. Other approaches use relationships between tasks as an auxiliary learning signal (Yao et al., 2019; Pal and Balasubramanian, 2019, e.g.), but cannot adapt to a new task by transforming their task representations. To address this, it is necessary to represent tasks in a transformable space. This can grant the ability to rapidly and precisely adapt behavior to a new task.

In this paper, we propose a new class of tasks based on this idea: meta-mappings, i.e. mappings between tasks (see below). Meta-mappings allow zero-shot performance of a new task based on its relationship to old tasks. To address the challenge of performing meta-mappings, we propose using architectures which essentially take a functional perspective on meta-learning, and exploit the idea of homoiconicity. (A homoiconic programming language is one in which programs in the language can be manipulated by programs in the language, just as data can.) By treating both data and task behaviors as functions, we can conceptually think of both data *and* learned task behaviors as transformable. This yields the ability to not only learn to solve new tasks, but to learn how to transform these solutions in response to changing task demands. We demonstrate that our architectures can flexibly remap their behavior to address the meta-mapping challenge. By allowing the network to recursively treat its task representations as data points, and transform them to produce new task

representations, our approach is able to achieve this flexibility parsimoniously. We suggest that approaches like ours will be key to building more intelligent and flexible deep learning systems.

## 2 META-MAPPING

We propose the meta-mapping challenge. We define a meta-mapping as a task, or mapping, that takes a task as an input, output, or both. These include mapping from tasks to language (explaining), mapping from language to tasks (following instructions), and mapping from tasks to tasks (adapting behavior). While the first two categories have been partially addressed in prior work (e.g. Hermann et al., 2017; Co-Reyes et al., 2019), the latter is more novel. (We discuss the relationship between our work and prior work in Section 7.) This adaptation can be cued in several ways, including examples of the mapping (after winning and losing at poker, try to lose at blackjack) or natural-language instructions ("try to lose at blackjack").

We argue that task-to-task meta-mappings are a useful way to think about human-like flexibility, because a great deal of our rapid adaptation is from a task to some variation on that task. For example, the task of playing go on a large board is closely related to the task of playing go on a small board. Humans can exploit this to immediately play well on a different board, but deep learning models generally have no way to achieve this. We can also adapt in much deeper ways, for example fundamentally altering our value function on a task, such as trying to lose, or trying to achieve some orthogonal goal. While meta-learning systems can rapidly learn a new task from a distribution of tasks they have experience with, this does not fully capture human flexibility. Given appropriate conditioning (see below), our architecture can use meta-mappings to adapt to substantial task alterations zero-shot, that is, without seeing a single example from the new task (Lake et al., 2017). Achieving this flexibility to meta-map to new tasks will be an important step toward more general intelligence – intelligence that is not limited to precisely the training tasks it has seen.

## 3 HOMOICONIC META-MAPPING (HOMM) ARCHITECTURE

To address these challenges, we propose HoMM architectures, composed of two components:

1. Input/output systems: domain specific encoders and decoders (vision, language, etc.) that map into a shared embedding space $Z$.
2. A meta-learning system that a) learns to embed tasks into the shared embedding space $Z$, b) learns to use these task embeddings to perform task-appropriate behavior, c) learns to embed meta-mappings into the same space, and d) learns to use these meta-mapping embeddings to transform basic task embeddings in a meta-mapping appropriate way.

These architectures are homoiconic because they have a completely shared $Z$ for individual data points, tasks, and meta-mappings. This parsimoniously allows for arbitrary mappings between these entities. In addition to basic tasks, the system can learn to perform meta-mappings to systematically alter task behavior based on relationships to other tasks. That is, it can transform task representations using the same components it uses to transform basic data points. (See also Appendix E.1.)

Without training on meta-mappings, of course, the system will not be able to execute them well. However, as we will show, if it is trained on a broad enough set of such mappings, it will be able to generalize to new instances drawn from the same meta-mapping distribution. For instances that fall outside its data distribution, or for optimal performance, it may require some retraining, however. This reflects the structure of human behavior – we are able to adapt rapidly when new knowledge is relatively consistent with our prior knowledge, but learning an entirely new paradigm (such as calculus for a new student) can be quite slow (cf. Kumaran et al., 2016; Botvinick et al., 2019).

More formally, we treat functions and data as entities of the same type. From this perspective, the data points that one function receives can themselves be functions[1]. The key insight is that then our architecture can transform data points[2] to perform basic tasks (as is standard in machine learning),

---

[1] Indeed, any data point can be represented as a constant function that outputs the data point.
[2] Where "data" is a quite flexible term. The approach is agnostic to whether the learning is supervised or reinforcement learning, whether inputs are images or natural language, etc.

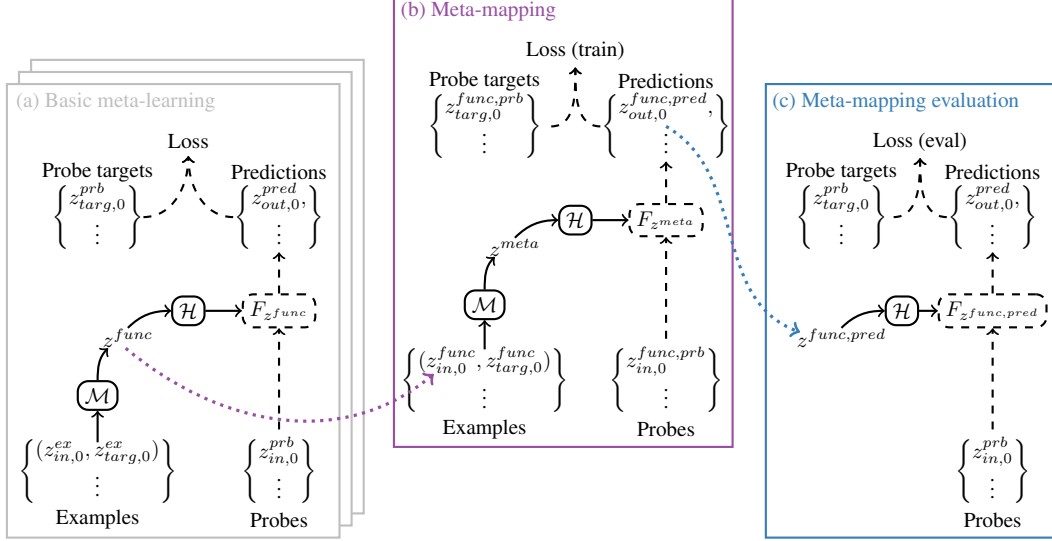

Figure 1: The HoMM architecture allows for transformations at different levels of abstraction. (a) For basic meta-learning a dataset consisting of (input embedding, output embedding) tuples is processed by the meta-network $\mathcal{M}$ to produce a function embedding $z^{func}$, which is processed by the hyper network $\mathcal{H}$ to parameterize a function $F_{z^{func}}$, which attempts to compute the transformation on probe inputs (used to encourage the system to generalize). However, our approach goes beyond basic meta-learning. The function embedding $z^{func}$ can then be seen as a single input or output at the next level of abstraction, when the same networks $\mathcal{M}$ and $\mathcal{H}$ are used to transform function embeddings based on examples of a meta-mapping (b). To evaluate meta-mapping performance, a probe embedding of a held-out function is transformed by the architecture to yield a predicted embedding for the transformed task. The performance of this predicted embedding is evaluated by moving back down a level of abstraction and evaluating on the actual target task (c). Because the function embedding is predicted by a transformation rather than from examples, new tasks can be performed zero-shot. ($M$ and $H$ are learnable deep networks, and $F_z$ is a deep network parameterized by $\mathcal{H}$ conditioned on function embedding $z$. Input and output encoders/decoders are omitted for simplicity. See the text and Appendix F.2 for details.)

but it can also transform these task functions to adapt to new tasks. This is related to the concepts of homoiconicity, defined above, and higher-order functions. Under this perspective, basic tasks and meta-mappings from task to task are really the same type of problem. The functions at one level of abstraction (the basic tasks) become inputs and outputs for higher-level functions at the next level of abstraction (meta-mapping between tasks).

Specifically, we embed each input, target, or mapping into a shared representational space $Z$. This means that single data points are embedded in the same space as the representation of a function or an entire dataset. Inputs are embedded by a deep network $\mathcal{I} : \text{input} \to Z$. Model outputs are decoded from $Z$ by $\mathcal{O} : Z \to \text{output}$. Target outputs are encoded by $\mathcal{T} : \text{targets} \to Z$.

Given this, the task of mapping inputs to outputs can be framed as trying to find a transformation of the representational space that takes the (embedded) inputs from the training set to embeddings that will decode to the target outputs. These transformations are performed by a system with the following components (see Fig. 1): $\mathcal{M} : \{(Z, Z), ...\} \to Z$ – the meta network, which collapses a dataset of (input embedding, target embedding) pairs to produce a single function embedding. $\mathcal{H} : Z \to \text{parameters}$ – the hyper network, which maps a function embedding to parameters. $F : Z \to Z$ – the transformation, implemented by a deep network parameterized by $\mathcal{H}$.

**Basic meta-learning:** To perform a basic task, input and target encoders ($\mathcal{I}$ and $\mathcal{T}$) are used to embed individual pairs from an example dataset $D_1$, to form a dataset of example (input, output) tuples (Fig. 1a). These examples are fed to $\mathcal{M}$, which produces a function embedding (via a deep

neural network, with several layers of parallel processing across examples, followed by an element-wise max across examples, and several more layers). This function embedding is mapped through the hyper network $\mathcal{H}$ to parameterize $F$, and then $F$ is used to process a dataset of embedded probe inputs, and $\mathcal{O}$ to map the resultant embeddings to outputs. This system can be trained end-to-end on target outputs for the probes. Having two distinct datasets forces generalization at the meta-learning level, in exactly the same way as standard meta-learning algorithms learn to generalize.

More explicitly, suppose we have a dataset of example input, target pairs ($D_1 = \{(x_0, y_0), ...\}$), and some input $x$ from a probe dataset $D_2$. The system would predict a corresponding output $\hat{y}$ as:

$$\hat{y} = \mathcal{O}\left(F_{z^{func}}\left(\mathcal{I}\left(x\right)\right)\right)$$

where $F_{z^{func}}$ is the meta-learner's representation for the function underlying the examples in $D_1$:

$$F_{z^{func}} \text{ is parameterized by } \mathcal{H}\left(z^{func}\right), \text{ where } z^{func} = \mathcal{M}\left(\{\left(\mathcal{I}\left(x_0\right), \mathcal{T}\left(y_0\right)\right), ...\}\right)$$

Then, given some loss function $\mathfrak{L}(y, \hat{y})$ defined on a single target output $y$ and an actual model output $\hat{y}$, we define our total loss computed on the probe dataset $D_2$ as:

$$\mathbb{E}_{(x,y) \in D_2}\left[\mathfrak{L}\left(y, \mathcal{O}\left(F_{D_1}\left(\mathcal{I}\left(x\right)\right)\right)\right)\right]$$

The system can then be trained end-to-end on this loss to adjust the weights of $\mathcal{T}, \mathcal{H}, \mathcal{M}, \mathcal{O}$, and $\mathcal{I}$. That is, the meta-learning training procedure optimizes the system for generalizing the function implied by the examples it sees, in order to respond appropriately to probe inputs. As in training a standard meta-learning model, these probes are held-out from the meta-network, but the labels are still used to encourage the system to meta-learn to respond appropriately to new inputs. Other data points are held-out entirely during training, and only used for evaluation, see Appendix A.1 for some detailed description. See Appendix F.2 for further details on the architecture and hyper-parameters.

**Meta-mapping:** The fundamental insight of our paper is to show how basic tasks and meta-mappings can be treated homogenously, by allowing the network to transform its task representations like data (see Fig. 1b,c). That is, we propose to transform a prior task representation in order to perform a related new task zero-shot. The relationship by which the prior task embedding should be transformed to perform the new task is specified by example tuples of (input, output) task embeddings. This is precisely analogous to how a basic task is specified in terms of (input, output) embeddings of data points, except that the embeddings now represent tasks that are being transformed, rather than individual data points.

Thus from the perspective of our architecture, learning a meta-mapping between tasks is exactly analogous to learning a basic task. Anything that is embedded in $Z$ can be transformed using the same system. Because tasks are embedded in $Z$ for basic meta-learning, this allows for meta-mappings using exactly the same $\mathcal{M}$ and $\mathcal{H}$ that we use for basic tasks. Just as we would take a set of paired embeddings of data points for a basic task, and use them to compute a function embedding for that task, we can take a set of paired function embeddings representing a meta-mapping, and use them to create an embedding for that meta-mapping. We can then use this meta-mapping embedding to transform the embedding of another basic task. We can thus behave zero-shot on a novel task based on its relationship to a prior task.

For example, suppose we have an embedding $z_{game1} \in Z$ for the task of playing some game, and we want to switch to trying to lose this game. We can generate a meta-mapping embedding $z_{\text{meta}} \in Z$ from examples of embeddings generated by the system when it is trying to win and lose various other games: $z_{\text{meta}} = \mathcal{M}\left(\{\left((z_{game2}, z_{game2,lose}), ...\}\right)\right)$. We can generate a new task embedding $\hat{z}_{game1,lose} \in Z$:

$$\hat{z}_{game1,lose} = F_{z_{\text{meta}}}(z_{game1}) \qquad \text{where } F_{z_{\text{meta}}} \text{ is parameterized by } \mathcal{H}\left(z_{\text{meta}}\right)$$

This $\hat{z}_{games1,lose}$ can be interpreted as the system's guess at a losing strategy for game 1. To train a meta-mapping, we minimize the $\ell_2$ loss in the latent space betwen this guessed embedding and the embedding of the target task[3]. Whether or not we have such a target embedding, we can

---

[3]The gradients do not update the example function embeddings, only the weights of $\mathcal{M}$ and $\mathcal{H}$, due to memory contraints. Allowing this might be useful in more complex applications.

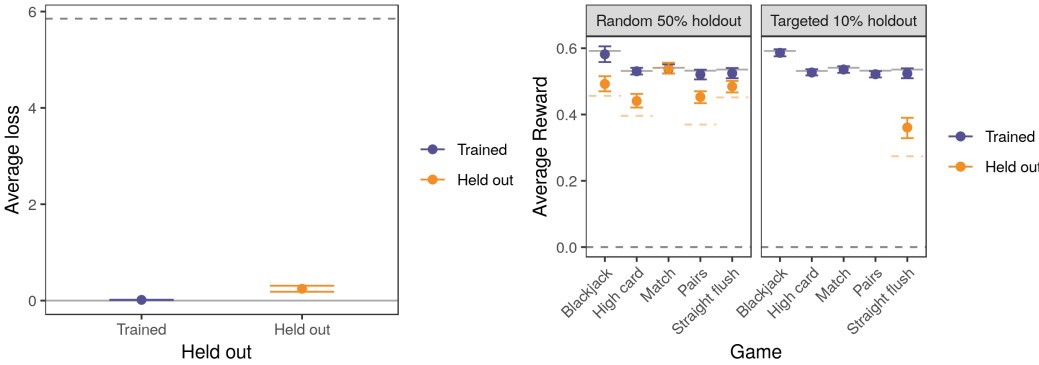

(a) The polynomials domain, Section 4.                    (b) The cards domain, Section 5.

Figure 2: The HoMM system succeeds at basic meta-learning, which is a necessary prerequisite for meta-mappings. (a) The polynomials domain, Section 4. The system successfully generalizes to held out polynomials. The solid line indicates optimal performance; the dashed line indicates untrained model performance. (b) The card games domain, Section 5. The system successfully generalizes to held out games, both when trained on a random sample of half the tasks, or when a targeted subset is held out. The gray dashed line indicates chance performance (otuputting the mean across all polynomials, or playing randomly), while the solid lines are optimal performance. The orange dashed lines shows performance on held-out tasks of playing the strategy from the most correlated trained task. The system generally exceeds this difficult baseline, thus showing deeper generalization than just memorizing strategies and picking the closest. Error-bars are bootstrap 95%-CIs, numerical values for plots can be found in Appendix G.

evaluate how well the system loses with this $\hat{z}_{game1,lose}$ strategy, by stepping back down a level of abstraction and actually having it play the game via this embedding (Fig. 1c). This is how we evaluate meta-mapping performance – evaluating the loss of transformed task embeddings on the respective target tasks. Note that this allows for zero-shot performance of the new task, because it simply requires transforming a prior task embedding.

Alternatively, we could map from language to a meta-mapping embedding, rather than inducing it from examples of the meta-mapping. This corresponds to the human ability to change behavior in response to instructions. The key feature of our architecture – the fact that tasks, data, and language are all embedded in a shared space – allows for substantial flexibility within a unified framework. Furthermore, our approach is parsimonious. Because it uses the same meta-learner for both basic tasks and meta-mappings, this increased flexibility does not require any added parameters.[4]

## 4 LEARNING MULTIVARIATE POLYNOMIALS

As a proof of concept, we first evaluated the system on the task of learning polynomials of degree $\leq 2$ in 4 variables (i.e. the task was to regress functions of the form $p : \mathbb{R}^4 \to \mathbb{R}$ where $p \in \mathcal{P}_2(\mathbb{R})$, though the model was given no prior inductive bias toward polynomial forms). For example, if $p(w, x, y, z) = x$, the model might see examples like $(-1, 1, 1, 1; 1)$ and $(0.7, 2.1, 1.3, -4; 2.1)$, and be evaluated on its output for points like $(-1, -1.3, 0.5, 0.3)$. This yields an infinite family of base-level tasks (the vector space of all such polynomials), as well as many families of meta-mappings over tasks (for example, multiplying polynomials by a constant, squaring them, or permuting their input variables). This allows us to not only examine the ability of the system to learn to learn polynomials from data, but also to adapt its learned representations in accordance with these meta-tasks. Details of the architecture and training can be found in Appendix F.

---

[4]At least in principle, in practice of course increasing network size might be more beneficial for HoMM architectures performing meta-mappings as well as basic tasks, compared to those performing only basic tasks.

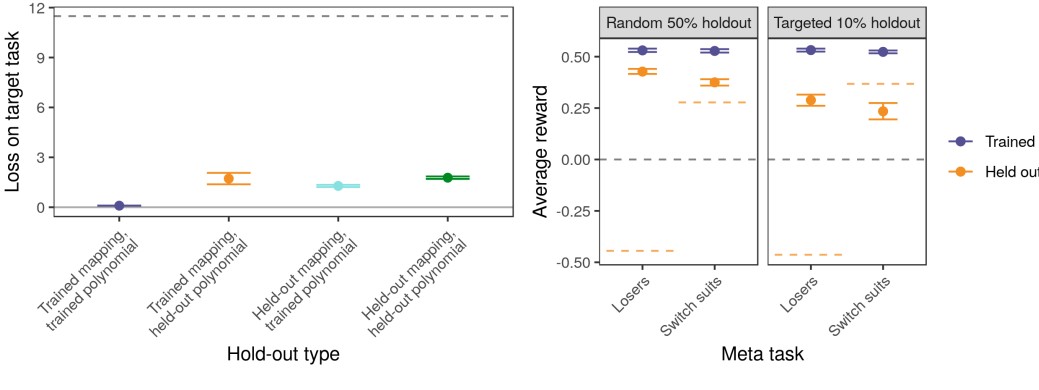

(a) The polynomials domain, Section 4.    (b) The cards domain, Section 5.

Figure 3: The HoMM architecture performs tasks zero-shot by meta-mappings. (a) The system generalizes to apply learned meta-mappings to new polynomials, and even to apply unseen meta-mappings. The plots show the loss produced when evaluating the mapped embedding on the target task. For example, if the initial polynomial is $p(x) = x + 1$, and the meta-task is "square," the loss would be evaluated by transforming the embedding of $p(x)$ and evaluating how well the mapped embedding regresses on to $p(x)^2 = x^2 + 2x + 1$. The results show that the system succeeds at applying meta-mappings it is trained on to held-out polynomials, as well as applying held-out meta-mappings to either trained or held-out polynomials. The solid line indicates optimal performance; the dashed line is untrained model performance. (b) The system generalizes to meta-mapping new tasks in the cards domain. The system is trained to do the meta-mappings shown here on a subset of its basic tasks, and is able to generalize these mappings to perform novel tasks zero-shot. For example, for the "losers" mapping, the sytem is trained to map games to their losers variants. When given a held-out game, it is able to apply the mapping to guess how to play the losing variation. This plot shows the reward produced by taking the mapped embedding and playing the targeted game. The gray dashed line indicates random performance, while the colored dashed lines indicate performance if the system did not alter its behavior in response to the meta-mapping. The system generally exceeds these baselines, although the switch-suits baseline is more difficult with the targeted holdout. Error-bars are bootstrap 95%-CIs.

**Basic meta-learning:**    First, we show that the system is able to achieve the basic goal of learning a held-out polynomial from a few data points in Fig. 2a (with good sample-efficiency, see Supp. Fig. 7).

**Zero-shot adaptation by meta-mapping (task $\rightarrow$ task):**    Furthermore, the system is able to perform meta-mappings over polynomials in order to flexibly reconfigure its behavior (Fig. 3a). We train the system to perform a variety of mappings, for example switch the first two inputs of the polynomial, add 3 to the polynomial, or square the polynomial. We then test its ability to generalize to held-out mappings from examples, for example a held-out input permutation, or an unseen additive shift. The system is both able to apply learned meta-mappings to held-out polynomials, and to apply held-out meta-mappings it has not been trained on, simply by seeing examples of the mapping.

## 5    A STOCHASTIC LEARNING SETTING: SIMPLE CARD GAMES

We next explored the setting of simple card games, where the agent is dealt a hand and must bet. There are three possible bets (including "don't bet"), and depending on the opponent's hand the agent either wins or loses the amount bet. This task doesn't require long term planning, but does incorporate some aspects of reinforcement learning, namely stochastic feedback on only the action chosen. We considered five games that are simplified analogs of various real card games (see Appendix F.1.2). We also considered several binary options that could be applied to the games, including trying to lose instead of trying to win, or switching which suit was more valuable. These are challenging manipulations, for instance trying to lose requires completely inverting a learned $Q$-function.

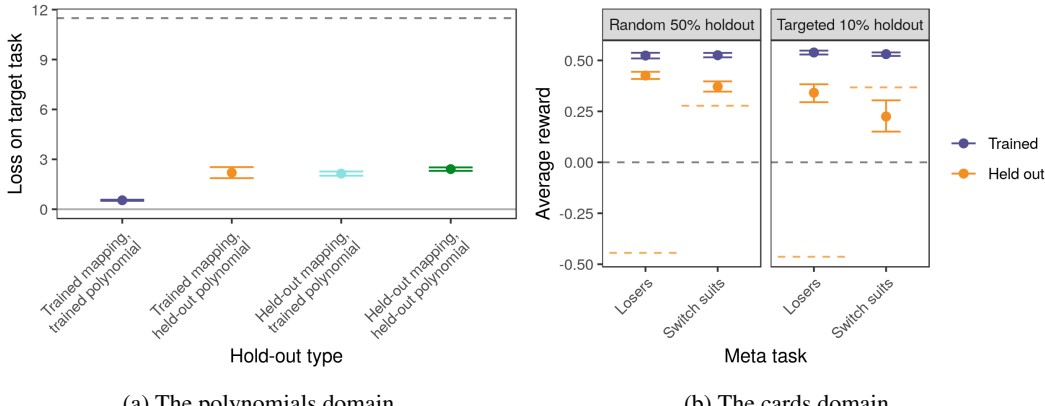

(a) The polynomials domain.

(b) The cards domain.

Figure 4: The HoMM system can perform meta-mappings from language cues rather than meta-mapping examples. Compare to Fig. 3, which shows the same results when using examples instead of language. (a) In the polynomials domain, language cues still lead to good performance, even on held-out tasks or held-out meta-mappings, although examples perform slightly better (Fig. 3a). (b) Similarly, in the cards domain, language cues perform well. Error-bars are bootstrap 95%-CIs.

In order to adapt the HoMM architecture, we made a very simple change. Instead of providing the system with (input, target) tuples to embed, we provided it with (state, action, reward) tuples, and trained it to predict rewards for each bet in each state. (A full RL framework is not strictly necessary here because there is no temporal aspect to the tasks; however, because the outcome is only observed for the action you take, it is not a standard supervised task.) The hand is explicitly provided to the network for each example, but which game is being played is implicitly captured in the training examples, without any explicit cues. That is, the system must learn to play directly from seeing a set of (state, action, reward) tuples which implicitly capture the structure and stochasticity of the game. We also trained the system to make meta-mappings, for example switching from trying to win a game to trying to lose. Details of the architecture and training can be found in Appendix F.

**Basic meta-learning:**  First, we show that the system is able to play a held-out game from examples in Fig. 2b. We compare two different hold-out sets: 1) train on half the tasks at random, or 2) specifically hold out all the "losers" variations of the "straight flush" game. In either of these cases, the meta-learning system achieves well above chance performance (0) at the held out tasks, although it is slightly worse at generalizing to the targeted hold out, despite having more training tasks in that case. Note that the sample complexity in terms of number of trained tasks is not that high, even training on 20 randomly selected tasks leads to good generalization to the held-out tasks. Furthermore, the task embeddings generated by the system are semantically organized, see Appendix D.

**Zero-shot adaptation by meta-mapping (task → task):**  Furthermore, the system is able to perform meta-mappings (mappings over tasks) in order to flexibly reconfigure its behavior. For example, if the system is trained to map games to their losers variations, it can generalize this mapping to a game it has not been trained to map, even if the source or target of that mapping is held out from training. In Fig. 3b we demonstrate this by taking the mapped embedding and evaluating the reward received by playing the targeted game with it. This task is more difficult than simply learning to play a held out game from examples, because the system will actually receive no examples of the target game (when it is held out). Furthermore, in the case of the losers mapping, leaving the strategy unchanged would produce a large negative reward, and chance performance would produce 0 reward, so the results are quite good.

## 6  AN EXTENSION VIA LANGUAGE

Language is fundamental to human flexibility. Often the examples of the meta-mapping are implicit in prior knowledge about the world that is cued by language. For example, "try to lose at go" does not give explicit examples of the "lose" meta-mapping, but rather relies on prior knowledge of what losing means. This is a much more efficient way to cue a known meta-mapping. In order to replicate this, we

trained the HoMM system with both meta-mappings based on examples, and meta-mappings based on language. In the language-based meta-mappings, a language input identifying the meta-mapping (but not the basic task to apply it to) is encoded by a language encoder, and then provided as the input to $\mathcal{H}$ (instead of an output from $\mathcal{M}$). The meta-mapping then proceeds as normal — $\mathcal{H}$ parameterizes $F$, which is used to transform the embedding of the input task to produce an embedding for the target.

This language-cued meta-mapping approach also yields good performance (Fig. 4). However, examples of the meta-mapping are slightly better, especially for meta-mappings not seen during training, presumably because examples provide a richer description. In the next section we show that using language to specify a meta-mapping performs better than using language to directly specify the target task.

### 6.1 WHY META-MAP FROM TASKS TO TASKS?

Why are meta-mappings between tasks useful? To answer this, we consider various ways of adapting to a new task in Figure 5 (based on results from the cards domain, Section 5). The standard meta-learning approach would be to adapt from seeing examples of the new task, but this requires going out and collecting data, which may be expensive and does not allow zero-shot adaptation. Alternatively, the system could perform the new task via a meta-mapping from a prior learned task, where the meta-mapping is either induced from examples of the meta-mapping, or from language. Finally, the system could perform a new task from language alone, if it is trained to map instructions to tasks. This is the more typical zero-shot paradigm (e.g. Romera-Paredes and Torr, 2015).

To address this latter possibility, we trained a version of the model where we included training the language system to produce embeddings for the basic tasks (while simultaneously training the system on all the other objectives, such as performing the tasks from examples, in order to provide the strongest possible structuring of the system's knowledge for the strongest possible comparison). We compare this model's performance at held-out tasks to that of systems learning from examples of the new task directly, or from meta-mapping, see Fig. 5.

These results demonstrate the advantage of meta-mapping. While learning from examples is still better given enough data, it requires potentially-expensive data collection and does not allow zero-shot adaptation. Performing the new task from a language description alone uses only the implicit knowledge in the model's weights, and likely because of this it does not generalize well to the difficult held-out tasks. Meta-mapping performs substantially better, while relying only on cached prior knowledge, viz. prior task-embedding(s) and a description of the meta-mapping (either in the form of examples or natural language). That is, meta-mapping allows zero-shot adaptation, like performing from language alone, but results in much better performance than language alone, by leveraging a richer description of the new task constructed using the system's knowledge of a prior task and the new task's relationship to it.

## 7 DISCUSSION

**Related work:** There has been a variety of past work on zero-shot learning, mostly based on natural-language descriptions of tasks. For example, Larochelle et al. (2008) considered the general problem of behaving in accordance with language instructions as simply asking a model to adapt its response when conditioned on different "instruction" inputs. Later work explored zero-shot classification based on only a natural language description of the target class (Socher et al., 2013; Romera-Paredes and Torr, 2015; Xian et al., 2018), or of a novel task in a language-conditioned RL system (Hermann et al., 2017). Some of this work has even exploited relationships between tasks as a learning signal (Oh et al., 2017). Other work has considered how similarity between tasks can be useful for generating representations for a new task (Pal and Balasubramanian, 2019), but without transforming task representations to do so. (Furthermore, similarity is less specific than an input-output mapping, since it does not specify *along which dimensions* two tasks are similar.) To our knowledge none of the prior work has proposed using meta-mapping approaches to adapt to new tasks by transforming task representations, nor has the prior work proposed a parsimonious homoiconic model which can perform these mappings.

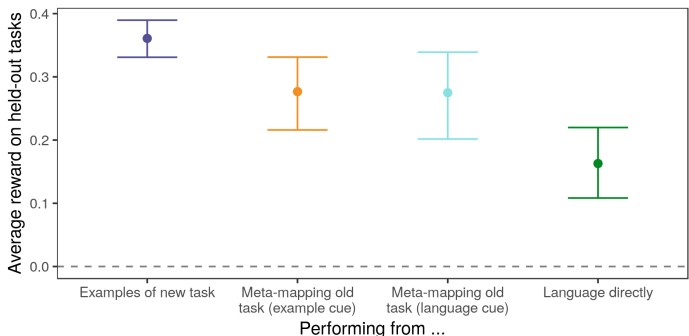

Figure 5: Comparison of a variety of methods for performing one of the 10% held-out tasks in the more difficult hold-out set in the cards domain. There are a number of ways the system could adapt to a new task: from seeing example of the new task, from hearing the new task described from language alone, or from leveraging its knowledge about prior tasks via meta-mappings (in this case, from the non-losers variations of the same games). The meta-mappings offer a happy medium between the other two alternatives – they only require cached knowledge of prior tasks, rather than needing to collect experience on the task before a policy can be derived, but they outperform a system that simply tries to construct the task embedding from a description alone. Language alone is not nearly as rich a cue as knowledge of how a new task relates to prior tasks.

Our work is an extrapolation from the rapidly-growing literature on meta-learning (e.g. Vinyals et al., 2016; Santoro et al., 2016; Finn et al., 2017; 2018; Stadie et al., 2018; Botvinick et al., 2019). It is also related to the literature on continual learning, or more generally tools for avoiding catastrophic interference based on changes to the architecture (e.g. Fernando et al., 2017; Rusu et al., 2016), loss (e.g. Kirkpatrick et al., 2016; Zenke et al., 2017; Aljundi et al., 2019), or external memory (e.g. Sprechmann et al., 2018). We also connect to a different perspective on continual learning in Appendix B. Recent work has also begun to blur the separation between these approaches, for example by meta-learning in an online setting (Finn et al., 2019). Our work is specifically inspired by the algorithms that attempt to have the system learn to adapt to a new task via activations rather than weight updates, either from examples (e.g. Wang et al., 2016; Duan et al., 2016), or a task input (e.g. Borsa et al., 2019).

Our architecture builds directly off of prior work on HyperNetworks (Ha et al., 2016) – networks which parameterize other networks – and other recent applications thereof, such as guessing parameters for a model to accelerate model search (e.g. Brock et al., 2018; Zhang et al., 2019), and meta-learning (Li et al., 2019; Rusu et al., 2019, e.g.). In particular, recent work in natural language processing has shown that having contextually generated parameters can allow for zero-shot task performance, assuming that a good representation for the novel task is given (Platanios et al., 2017) – in their work this representation was evident from the factorial structure of translating between many pairs of languages. Our work is also related to the longer history of work on different time-scales of weight adaptation (Hinton and Plaut, 1982; Kumaran et al., 2016) that has more recently been applied to meta-learning contexts (e.g. Ba et al., 2016; Munkhdalai and Yu, 2017; Garnelo et al., 2018) and continual learning (Hu et al., 2019, e.g.). It is more abstractly related to work on learning to propose architectures (e.g. Zoph and Le, 2016; Cao et al., 2019), and to models that learn to select and compose skills to apply to new tasks (e.g. Andreas et al., 2016b;a; Tessler et al., 2016; Reed and de Freitas, 2015; Chang et al., 2019). In particular, some of the work in domains like visual question answering has explicitly explored the idea of building a classifier conditioned on a question (Andreas et al., 2016b; 2017), which is related to one of the possible computational paths through our architecture. Work in model-based reinforcement learning has also partly addressed how to transfer knowledge between different reward functions (e.g. Laroche and Barlier, 2017); our approach is more general. Indeed, our insights could be combined with model-based approaches, for example our approach could be used to adapt a task embedding, which would then be used by a learned planning model.

There has also been other recent interest in task (or function) embeddings. Achille et al. (Achille et al., 2019) recently proposed computing embeddings for visual tasks from the Fisher information of the parameters in a model partly tuned on the task. They show that this captures some interesting properties of the tasks, including some types of semantic relationships, and can help identify models that can perform well on a task. Rusu and colleagues recently suggested a similar meta-learning framework where latent codes are computed for a task which can be decoded to a distribution over parameters (Rusu et al., 2019). Other recent work has tried to learn representations for skills (e.g. Eysenbach et al., 2019) or tasks (Hsu et al., 2019, e.g.) for exploration and representation learning. Our perspective can be seen as a generalization of these that allows for remapping of behavior via meta-tasks. While there have been other approaches to zero-shot task performance, to the best of our knowledge none of the prior work has explored zero-shot performance of a task via meta-mappings.

**Future Directions:** We think that the general perspective of considering meta-mappings will yield many fruitful future directions. We hope that our work will inspire more exploration of behavioral adaptation, in areas beyond the simple domains we considered here. To this end, we suggest the creation of meta-learning datasets which include information not only about tasks, but about the relationships between them. For example, reinforcement learning tasks which involve executing instructions (e.g. Hermann et al., 2017; Co-Reyes et al., 2019) can be usefully interpreted from this perspective. Furthermore, we think our work provides a novel perspective on the types of flexibility that human intelligence exhibits, and thus hope that it may have implications for cognitive science.

We do not necessarily believe that the particular architecture we have suggested is the best architecture for addressing these problems, although it has a number of desirable characteristics. However, the modularization of the architecture makes it easy to modify. (We compare some variations in Appendix E.) For example, although we only considered task networks $F$ that are feed-forward and of a fixed depth, this could be replaced with a recurrent architecture to allow more adaptive computation, or even a more complex architecture (e.g. Reed and de Freitas, 2015; Graves et al., 2016). Our work also opens the possibility of doing unsupervised learning over function representations for further learning, which relates to long-standing ideas in cognitive science about how humans represent knowledge (Clark and Karmiloff-Smith, 1993).

## 8 CONCLUSIONS

We've highlighted a new type of flexibility in the form of meta-mapping between tasks. Meta-mapping can produce zero-shot performance on novel tasks, based on their relationship to old tasks. This is a key aspect of human flexibility. The meta-mapping perspective suggests that this flexibility requires representing tasks in a transformable way. To address this, our Homoiconic Meta-Mapping (HoMM) approach explicitly represents tasks by function embeddings, and derives the computation from these embeddings via a HyperNetwork. This allows for our key insight: by embedding tasks and individual data points in a shared latent space, the same meta-learning architecture can be used both to transform data for basic tasks, and to transform task embeddings to adapt to task variations. This approach is parsiomious, because it uses the same networks for basic- and meta-mappings. Perhaps because of this, the requisite meta-task sample complexity is small; we showed generalization to unseen meta-mappings after training on only 20 meta-mappings. That is, we are able to achieve zero-shot performance on a novel task based on a held-out relationship between tasks. Furthermore, the meta-mapping approach yields better task performance than a zero-shot approach based on performing the task based on just a natural language description. This is a step closer to human-level flexibility.

We see our proposal as a logical progression from the fundamental idea of meta-learning – that there is a continuum between data and tasks. This naturally leads to the idea of manipulating task representations just like we manipulate data. We've shown that this approach yields considerable flexibility, most importantly the meta-mapping ability to adapt zero-shot to a new task. We hope that these results will lead to the development of more powerful and flexible deep-learning models.

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

# A    CLARIFYING META-MAPPING

## A.1    CLARIFYING HOLD-OUTS

There are several distinct types of hold-outs in the basic training of our architecture:

1. On each basic task, some of the data ($D_1$) is fed to the meta-network as examples $\mathcal{M}$ while some probes ($D_2$) is held out. This encourages the model to actually infer the underlying function, rather than just memorizing the examples. However, the probe labels are still used in training the system end-to-end, in order to encourage it to generalize.

2. There are also truly held-out tasks that the system has never seen in training. These are the held-out tasks that we evaluate on at the end of training and that are plotted in the "Held out" sections in the main plots.

This applies analogously to the meta-mappings: each time a meta-mapping is trained, some basic tasks are used as examples while others are held out to encourage generalization. There are also meta-mappings which have never been encountered during training, which we evaluate on at the end of training, those are the meta-mappings which are plotted in the "held out" section in the relevant plots. We also evaluate the old (and new) meta-mappings on the new basic tasks that have never been trained.

## A.2    A DEFINITIONAL NOTE

When we discussed meta-mappings in the main task, we equivocated between tasks and behaviors for the sake of brevity. For a perfect model, this is somewhat justifiable, because each task will have a corresponding optimal behavior, and the sytem's embedding of the task will be precisely the embedding which produces this optimal behavior. However, behavior-irrelevant details of the task, like the color of the board, may not be embedded, so this should not really be thought of as a task-to-task mapping. This problem is exacerbated when the system is imperfect, e.g. during learning. It is thus more precise to distinguish between a ground-truth meta-mapping, which maps tasks to tasks, and the computational approach to achieving that meta-mapping, which really maps between representations which combine both task and behavior.

## B CONTINUAL LEARNING

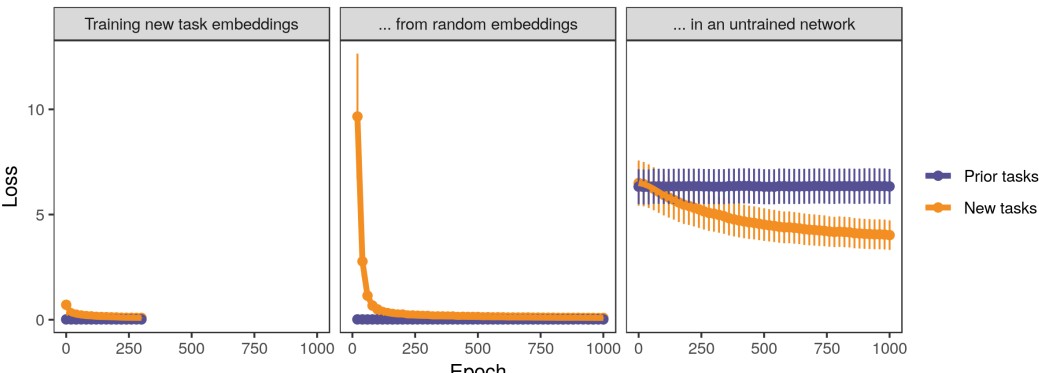

Figure 6: Once the meta-learning system has been trained on a distribution of prior tasks, its performance on new tasks can be tuned by caching its guessed embeddings for the tasks and then optimizing those, thus avoiding any possibility of interfering with performance on prior tasks. Starting from random embeddings in the trained model results in slower convergence, while in an untrained model the embeddings cannot be optimized well. Error-bars are bootstrap 95%-CIs.

**Continual learning:** Although the meta-learning approach is effective for rapidly adapting to a new task, it is unreasonable to think that our system must consider every example it has seen at each inference step. We would like to be able to store our knowledge more efficiently, and allow for further refinement. Furthermore, we would like the system to be able to adapt to new tasks (for which its guessed solution isn't perfect) without catastrophically interfering with prior tasks (McCloskey and Cohen, 1989).

A very simple solution to these problems is naturally suggested by our architecture. Specifically, task embeddings can be cached so that they don't have to be regenerated at each inference step. This also allows optimization of these embeddings without altering the other parameters in the architecture, thus allowing fine-tuning on a task without seeing more examples, and without interfering with performance on any other task (cf. Rumelhart and Todd, 1993; Lampinen and McClelland, 2018). This is like the procedure of Rusu et al. (2019), except considered across episodes. That is, we can see the meta-learning step as a "warm start" for an optimization procedure over embeddings that are cached in memory (cf. Kumaran et al., 2016). While this is not a traditional continual learning perspective, we think it provides an interesting perspective on the issue. It might in fact be much more memory-efficient to store an embedding per task, compared to storing an extra "importance" parameter for every parameter in our model, as in e.g. elastic weight consolidation (Kirkpatrick et al., 2016). It also provides a stronger guarantee of non-interference.

To test this idea, we pre-trained the system on 100 polynomial tasks, and then introduced 100 new tasks. We trained on these new tasks by starting from the meta-network's "guess" at the correct task embedding, and then optimizing this embedding without altering the other parameters. The results are shown in Fig. 6. The meta-network embeddings offer good immediate performance, and substantially accelerate the optimization process, compared to a randomly-initialized embedding (see supp. Fig. 10 for a more direct comparison). Furthermore, this ability to learn is due to training, not simply the expressiveness of the architecture, as is shown by attempting the same with an untrained network.

## C  SUPPLEMENTAL FIGURES

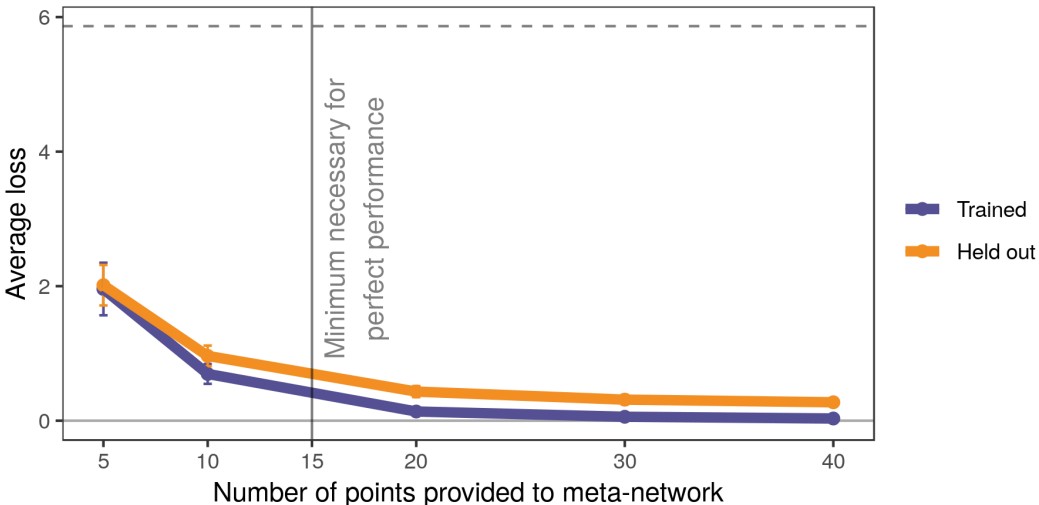

Figure 7: The system is able to infer polynomials from only seeing a few data points (i.e. evaluations of the polynomial), despite the fact that during training it always saw 50. A minimum of 15 random points is needed to correctly infer polynomials without prior knowledge of the polynomial distribution, but the system is performing well below this value, and quite well above it, although it continues to refine its estimates slightly when given more data.

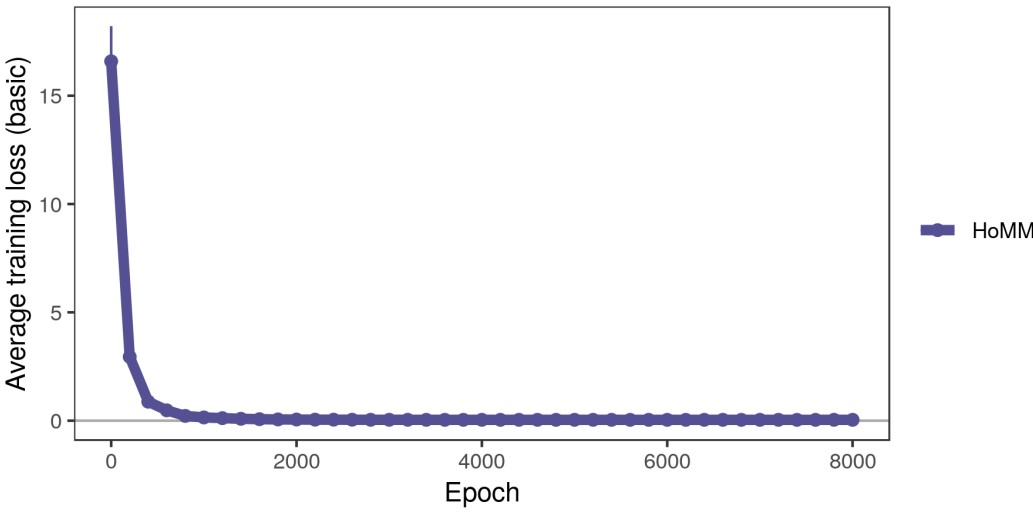

Figure 8: Learning curves for basic regression in the polynomials domain.

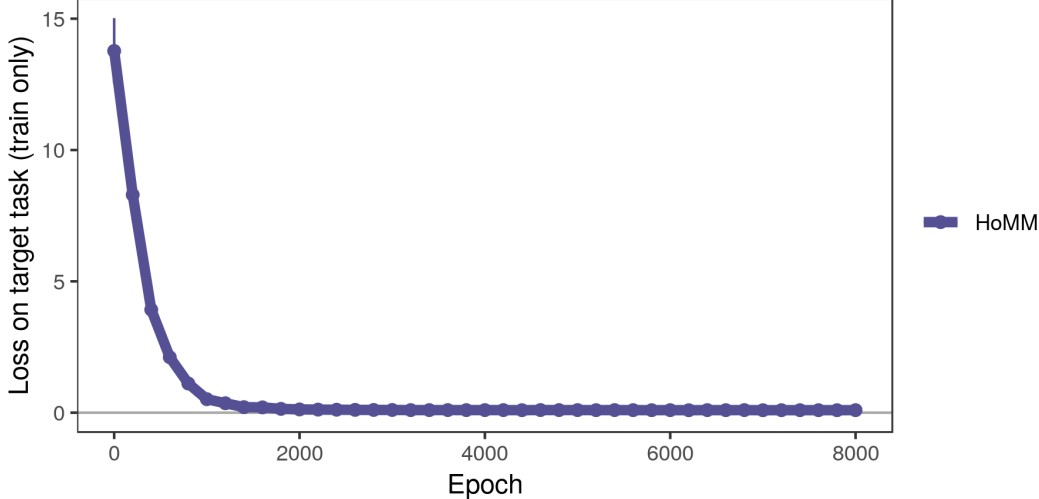

Figure 9: Learning curves for meta-mappings in the polynomials domain. Although the results seem to be leveling off at the end, we found that generalization performance was slightly increasing or stable in this region, which may have interesting implications about the structure of these tasks (Lampinen and Ganguli, 2019).

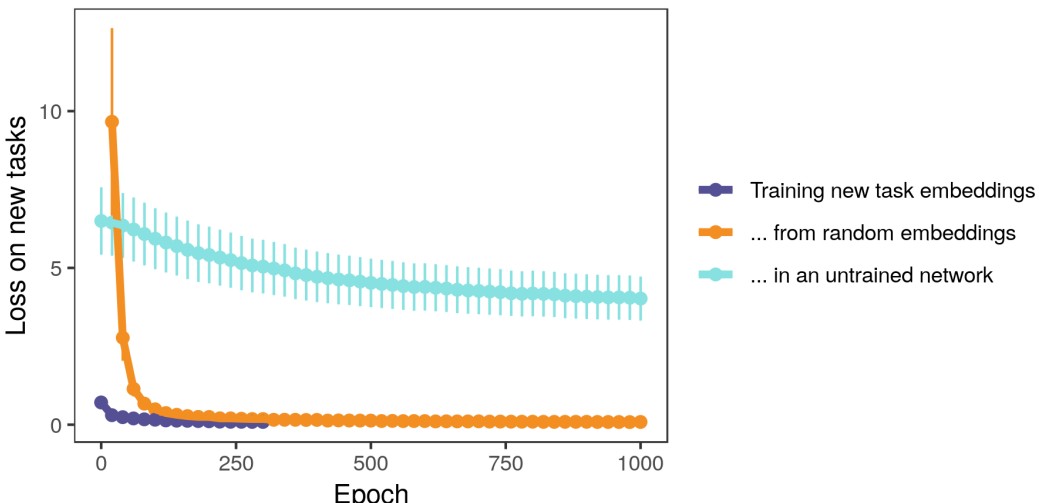

Figure 10: Continual learning in the polynomials domain: a more direct comparison. Once the meta-learning system has been trained on a distribution of prior tasks, its performance on new tasks can be tuned by caching its guessed embeddings for the tasks and then optimizing those, thus avoiding any possibility of interfering with performance on prior tasks. Starting with the guessed embedding substantially speeds-up the process compared to a randomly-initialized embedding. Furthermore, this ability to learn is due to training, not simply the expressiveness of the architecture, as is shown by attempting the same with an untrained network.

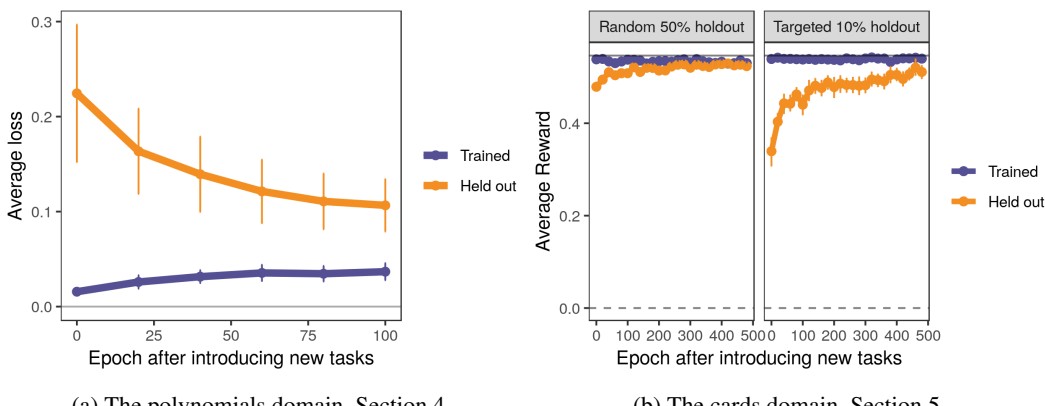

(a) The polynomials domain, Section 4.

(b) The cards domain, Section 5.

Figure 11: Integrating new tasks into the system by training all parameters results in some initial interference with prior tasks (even with replay), suggesting that an approach like the continual learning-approach may be useful.

# D  CARD GAME $t$-SNE

We performed $t$-SNE (Laurens van der Maaten and Hinton, 2008) on the task embeddings of the system at the end of learning the card game tasks, to evaluate the organization of knowledge in the network. In Fig. 12 we show these embeddings for just the basic tasks. The embeddings show systematic grouping by game attributes. In Fig. 13 we show the embeddings of the meta and basic tasks, showing the organization of the meta-tasks by type.

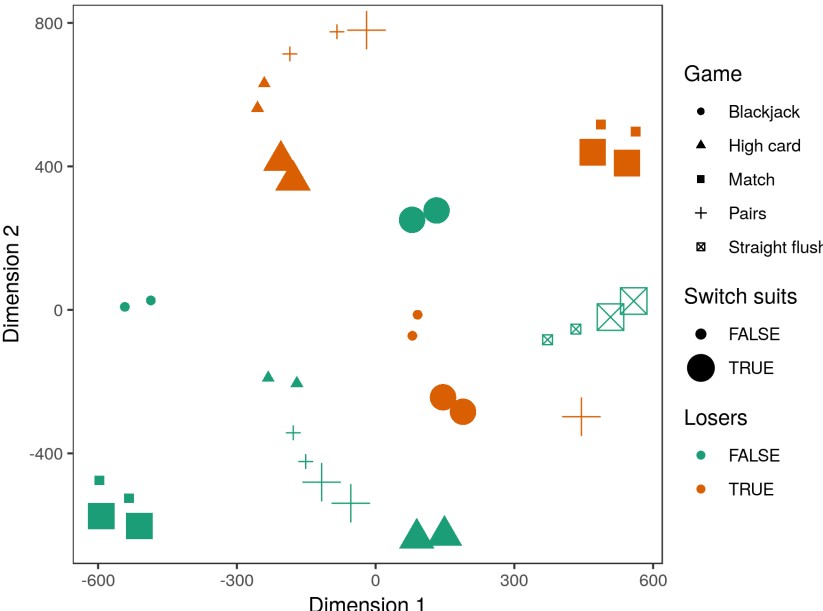

Figure 12: $t$-SNE embedding of the function embeddings the system learned for the basic card game tasks. (Note that the pairs of nearby embeddings differ in the "suits rule" attribute, discussed in Appendix F.1.2.)

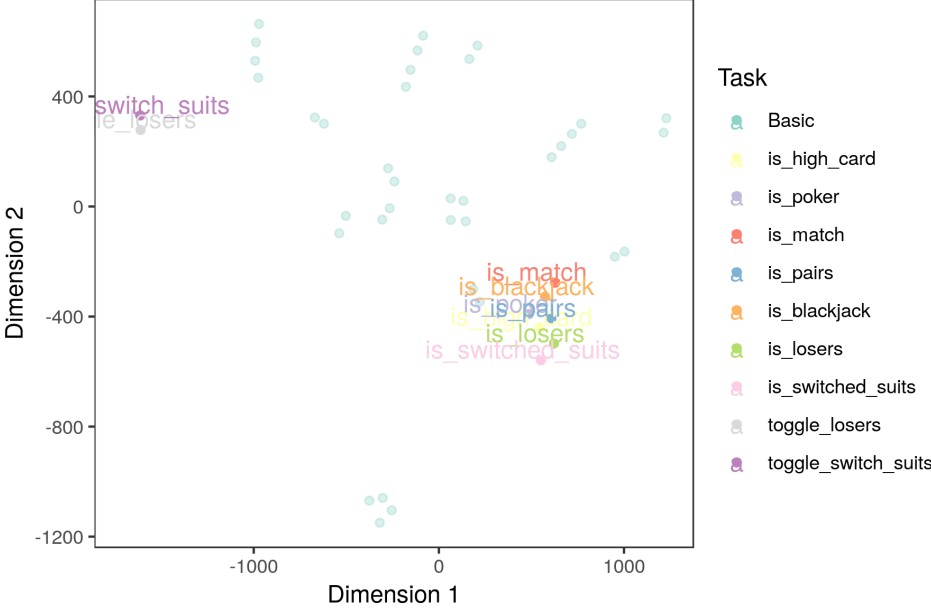

Figure 13: $t$-SNE embedding of the function embeddings the system learned for the meta tasks (basic tasks are included in the background).

# E    ARCHITECTURE EXPERIMENTS

In this section we consider a few variations of the architecture, to justify the choices made in the paper.

## E.1    SHARED $Z$ VS. SEPARATE TASK-EMBEDDING AND DATA-EMBEDDING SPACE

Instead of having a shared $Z$ where data and tasks are embedded, why not have a separate embedding space for data, tasks, and so on? There are a few conceptual reason why we chose to have a shared $Z$, including its greater parameter efficiency, the fact that humans seem to represent our conscious knowledge of different kinds in a shared space (Baars, 2005), and the fact that this representation could allow for zero-shot adaptation to new computational pathways through the latent space, analogously to the zero-shot language translation results reported by Johnson and colleagues (Johnson et al., 2016). In this section, we further show that training with a separate task encoding space worsens performance, see Fig. 14. This seems to primarily be due to the fact that learning in the shared $Z$ accelerates and de-noises the learning process, see Fig. 15. (It's therefore worth noting that running this model for longer could result in convergence to the same asymptotic generalization performance.)

## E.2    HYPER NETWORK VS. CONDITIONED TASK NETWORK

Instead of having the task network $F$ parameterized by the hyper network $\mathcal{H}$, we could simply have a task network with learned weights which takes a task embedding as another input. Here, we show that this architecture fails to learn the meta-mapping tasks, although it can successfully perform the basic tasks. We suggest that this is because it is harder for this architecture to prevent interference between the comparatively larger number of basic tasks and the smaller number of meta-tasks. While it might be possible to succeed with this architecture, it was more difficult in the hyper-parameter space we searched.

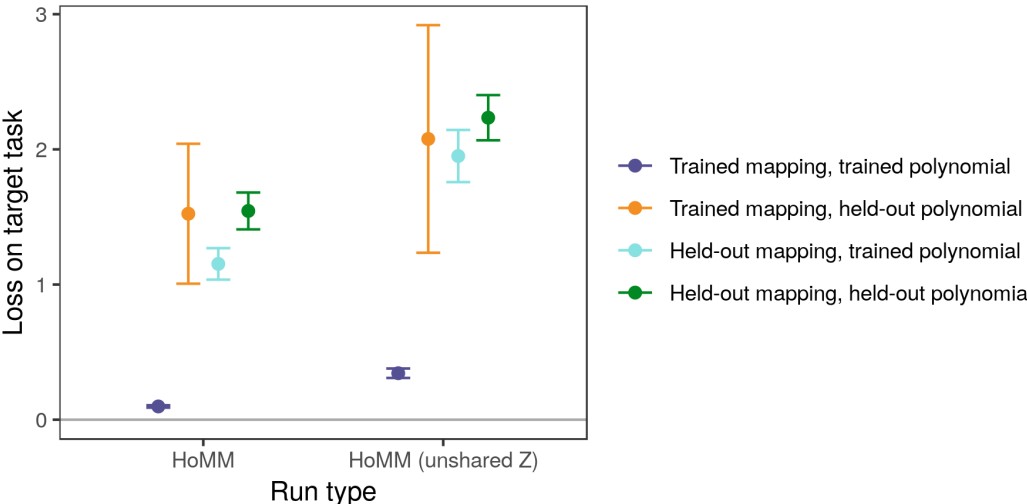

Figure 14: Having a separate embedding space for tasks results in worse performance on meta-mappings. (Results are from only 1 run.)

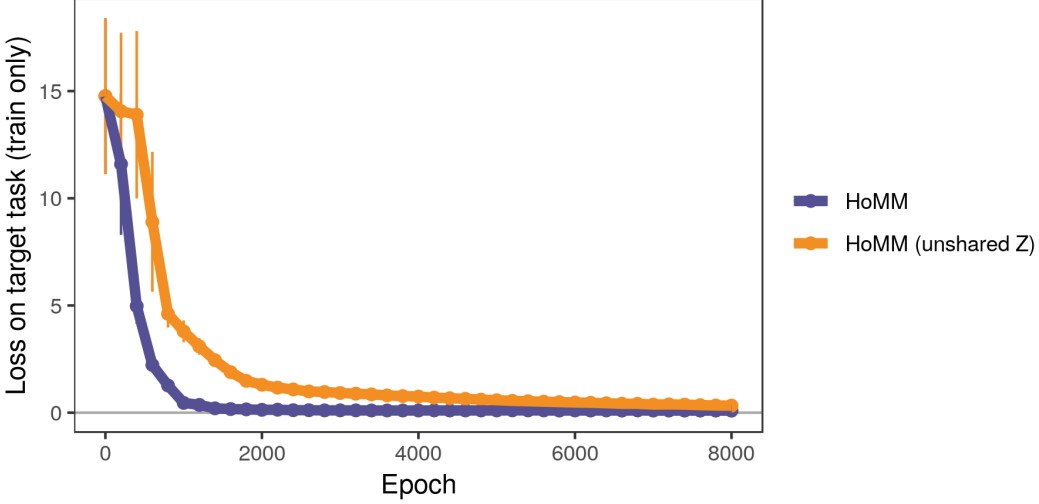

Figure 15: Having a separate embedding space for tasks results in noisier, slower learning of meta-mappings. (Results are from only 1 run.)

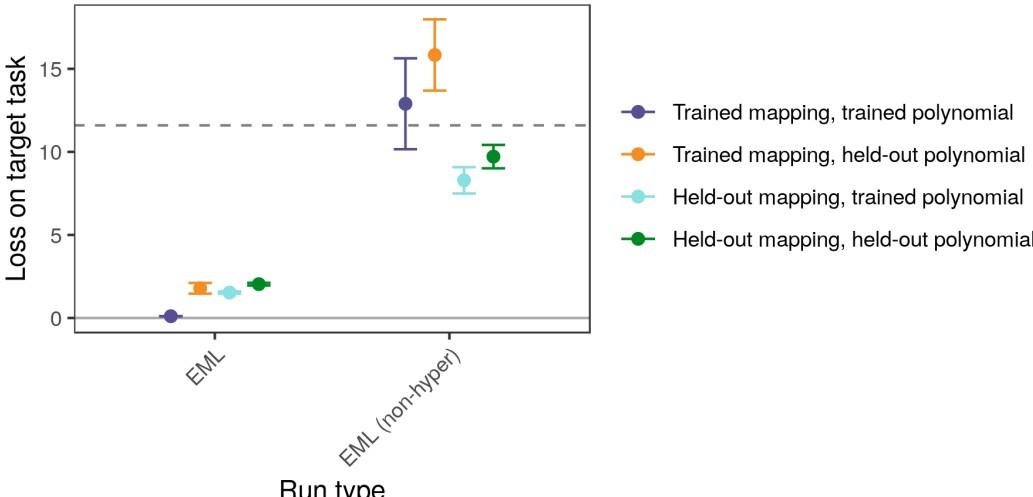

Figure 16: Conditioning the task network on the task embedding, rather than parameterizing it via a hyper network causes it to fail at the meta-mapping tasks. Results are from only 2 runs.

## F   DETAILED METHODS

### F.1   DATASETS

#### F.1.1   POLYNOMIALS

We randomly sampled the train and test polynomials as follows:

1. Sample the number of relevant variables ($k$) uniformly at random from 0 (i.e. a constant) to the total number of variables.
2. Sample the subset of $k$ variables that are relevant from all the variables.
3. For each term combining the relevant variables (including the intercept), include the term with probability 0.5. If so give it a random coefficient drawn from $\mathcal{N}(0, 2.5)$.

The data points on which these polynomials were evaluated were sampled uniformly from $[-1, 1]$ independently for each variable, and for each polynomial. The datasets were resampled every 50 epochs of training.

**Meta-tasks:** For meta-tasks, we trained the network on 6 task-embedding classification tasks:

- Classifying polynomials as constant/non-constant.
- Classifying polynomials as zero/non-zero intercept.
- For each variable, identifying whether that variable was relevant to the polynomial.

We trained on 20 meta-mapping tasks, and held out 16 related meta-mappings.

- Squaring polynomials (where applicable).
- Adding a constant (trained: -3, -1, 1, 3, held-out: 2, -2).
- Multiplying by a constant (trained: -3, -1, 3, held-out: 2, -2).
- Permuting inputs (trained: 1320, 1302, 3201, 2103, 3102, 0132, 2031, 3210, 2301, 1203, 1023, 2310, held-out: 0312, 0213, 0321, 3012, 1230, 1032, 3021, 0231, 0123, 3120, 2130, 2013).

**Language:** We encoded the meta-tasks in language by sequences as follows:

- Classifying polynomials as constant/non-constant: [``is'', ``constant'']
- Classifying polynomials as zero/non-zero intercept: [``is'', ``intercept_nonzero'']
- For each variable, identifying whether that variable was relevant to the polynomial: [``is'', <variable-name>, ``relevant'']
- Squaring polynomials: [``square'']
- Adding a constant: [``add'', <value>]
- Multiplying by a constant: [``multiply'', <value>]
- Permuting inputs:

  [``permute'', <variable-name>, <variable-name>, <variable-name>, <variable-name>]

All sequences were front-padded with "<PAD>" to the length of the longest sequence.

### F.1.2  CARD GAMES

Our card games were played with two suits, and 4 values per suit. In our setup, each hand in a game has a win probability (proportional to how it ranks against all other possible hands). The agent is dealt a hand, and then has to choose to bet 0, 1, or 2 (the three actions it has available). We considered a variety of games which depend on different features of the hand:

- **High card:** Highest card wins.
- **Pairs** Same as high card, except pairs are more valuable, and same suit pairs are even more valuable.
- **Straight flush:** Most valuable is adjacent numbers in same suit, i.e. 4 and 3 in most valuable suit wins every time (royal flush).
- **Match:** the hand with cards that differ least in value (suit counts as 0.5 pt difference) wins.
- **Blackjack:** The hand's value increases with the sum of the cards until it crosses 5, at which point the player "goes bust," and the value becomes negative.

We also considered three binary attributes that could be altered to produce variants of these games:

- **Losers:** Try to lose instead of winning! Reverses the ranking of hands.
- **Suits rule:** Instead of suits being less important than values, they are more important (essentially flipping the role of suit and value in most games).
- **Switch suit:** Switches which of the suits is more valuable.

Any combination of these options can be applied to any of the 5 games, yielding 40 possible games. The systems were trained with the full 40 possible games, but after training we discovered that the "suits rule" option does not substantially alter the games we chose (in the sense that the probability of a hand winning in the two variants of a game is very highly correlated), so we have omitted it from our analyses.

**Meta-tasks:** For meta-tasks, we gave the network 8 task-embedding classification tasks (one-vs-all classification of each of the 5 game types, and of each of the 3 attributes), and 3 meta-mapping tasks (each of the 3 attributes).

**Language:** We encoded the meta-tasks in language by sequences of the form [``toggle'', <attribute-name>] for the meta-mapping tasks, and [``is'', <attribute-or-game-name>].

## F.2 MODEL & TRAINING

**Basic task operation:**

1. A training dataset $D_1$ of (input, target) pairs is embedded by $\mathcal{I}$ and $\mathcal{T}$ to produce a set of paired embeddings. Another set of (possibly unlabeled) inputs $D_2$ is provided and embedded.

2. The meta network $\mathcal{M}$ maps the set of embedded (input, target) pairs to a function embedding.

3. The hyper network $\mathcal{H}$ maps the function embedding to parameters for $F$, which is used to transform the second set of inputs to a set of output embeddings.

4. The output embeddings are decoded by $\mathcal{O}$ to produce a set of outputs.

5. The system is trained end-to-end to minimize the loss on these outputs.

The model is trained to minimize

$$\mathbb{E}_{(x,y)\in D_2} \left[ \mathfrak{L} \left( y, \mathcal{O} \left( F_{D_1} \left( \mathcal{I} \left( x \right) \right) \right) \right) \right]$$

where $F_{D_1}$ is the transformation the meta-learner guesses for the training dataset $D_1$:

$$F_{D_1} \text{ is parameterized by } \mathcal{H} \left( \mathcal{M} \left( \{ \left( \mathcal{I} \left( x_i \right), \mathcal{T} \left( y_i \right) \right) \text{ for } \left( x_i, y_i \right) \in D_1 \} \right) \right)$$

**Meta-task operation:**

1. A meta-dataset of (source-task-embedding, target-task-embedding) pairs, $D_1$, is collected. Another dataset $D_2$ (possibly only source tasks) is provided. (All embeddings included in $D_1$ during training are for basic tasks that have themselves been trained, to ensure that there is useful signal. During evaluation, the embeddings in $D_1$ are for tasks that have been trained on, but those in $D_2$ may be new.

2. The meta network $\mathcal{M}$ maps this set of (source, target) task-embedding pairs to a function embedding.

3. The hyper network $\mathcal{H}$ maps the function embedding to parameters for $F$, which is used to transform the second set of inputs to a set of output embeddings.

4. The system is trained to minimize $\ell_2$ loss between these mapped embeddings and the target embeddings.

The model is trained to minimize

$$\mathbb{E}_{(z_{source}, z_{target})\in D_2} \left[ \mathfrak{L} \left( z_{target}, F_{D_1} \left( \mathcal{I} \left( z_{source} \right) \right) \right) \right]$$

where $\mathfrak{L}$ is $\ell_2$ loss, and $F_{D_1}$ is the transformation the meta-learner guesses for the training dataset $D_1$:

$$F_{D_1} \text{ is parameterized by } \mathcal{H} \left( \mathcal{M} \left( \{ \left( \left( z_{source}, z_{target} \right) \in D_1 \} \right) \right) \right)$$

Note that there are three kinds of hold-out in the training of this system, see Section A.1.

**Language-cued meta-tasks:** The procedure is analogous to the meta-tasks from examples, except that the input to $\mathcal{H}$ is the embedding of the language input, rather than the output of $\mathcal{M}$. The systems that were trained from language were also trained with the example-based meta-tasks.

### F.2.1 DETAILED HYPER-PARAMETERS

See table 1 for detailed architectural description and hyperparameters for each experiment. Hyperparameters were generally found by a heuristic search, where mostly only the optimizer, learning rate annealing schedule, and number of training epochs were varied, not the architectural parameters. Some of the parameters take the values they do for fairly arbitrary reasons, e.g. the continual learning experiments were run with the current polynomial hyperparameters before the hyperparameter search for the polynomial data was complete, so some parameters are altered between these.

| | Polynomials | Continual learning | Cards |
|---|---|---|---|
| $Z$-dimension | 512 | 512 | 512 |
| $\mathcal{I}$ num. layers | | 3 | |
| $\mathcal{I}$ num. hidden units | | 64 | |
| $\mathcal{L}$ architecture | 2-layer LSTM + 2 fully-connected | - | 1-layer LSTM + 2 fully-connected |
| $\mathcal{L}$ num. hidden units | 512 | - | 512 |
| $\mathcal{O}$ num. layers | 1 | 1 | 3 |
| $\mathcal{O}$ num. hidden units | - | - | 512 |
| $\mathcal{T}$ num. layers | | 1 | |
| $\mathcal{M}$ architecture | | 2 layers per-datum, max pool across, 2 layers | |
| $\mathcal{H}$ architecture | | 4 layers | |
| $\mathcal{M}, \mathcal{H}$ num. hidden units | | 512 | |
| $F$ architecture | | 4 layers | |
| $F$ num. hidden units | | 64 | |
| Nonlinearities | | Leaky ReLU in most places, except no non-linearity at final layer of $\mathcal{T}$, $\mathcal{M}$, $\mathcal{L}$, $F$, and sigmoid for meta-classification outputs. | |
| Main loss | | $\ell_2$ for main task & meta-mapping, cross-entropy for meta-classification. | |
| Optimizer | Adam | RMSProp | RMSProp |
| Learning rate (base) | $3 \cdot 10^{-5}$ | $1 \cdot 10^{-4}$ | $1 \cdot 10^{-4}$ |
| Learning rate (meta) | $1 \cdot 10^{-5}$ | - | $1 \cdot 10^{-4}$ |
| L.R. decay rate (base) | $*0.85$ | $*0.85$ | $*0.85$ |
| L.R. decay rate (meta) | $*0.85$ | - | $*0.9$ |
| L.R. min (base) | | $3 \cdot 10^{-8}$ | |
| L.R. min (meta) | $1 \cdot 10^{-7}$ | - | $3 \cdot 10^{-7}$ |
| L.R. decays every | | 100 epochs if above min. | |
| Cached embedding L.R. | - | $1 \cdot 10^{-3}$ | - |
| Num. training epochs | 4000 | 3000 | 40000 |
| Num. runs | 5 | 5 | 10 |
| Num. base tasks (eval) | 60 | 100 | 36 or 20 |
| Num. base tasks (training) | 1200 (= 60 * 20) | 100 | 36 or 20 |
| Num. meta classifications | 6 | - | 8 |
| Num. meta mappings | 20 | - | 3 |
| Num. new base tasks | 40 | 30 | 4 or 20 |
| Num. new meta mappings | 16 | - | 0 |
| Num. new meta classifications | | 0 | |
| Base dataset size | | 1024 | |
| Base datasets refreshed | | Every 50 epochs | |
| $\mathcal{M}$ batch size | 50 | 128 | 768 |

Table 1: Detailed hyperparameter specification for different experiments. A "-" indicates a parameter that does not apply to that experiment. Where only one value is given, it applied to all the experiments discussed. As a reminder: the shared representational space is denoted by $Z$. Input encoder: $\mathcal{I}$ : input $\rightarrow Z$. Output decoder $\mathcal{O}$ : $Z \rightarrow$ output. Target encoder $\mathcal{T}$ : targets $\rightarrow Z$. Meta-network $\mathcal{M}$ : $\{(Z, Z), ...\} \rightarrow Z$ – takes a set of (input embedding, target embedding) pairs and produces a function embedding. Hyper-network $\langle$ : $Z \rightarrow$ parameters – takes a function embedding and produces a set of parameters. Task network $F$ : $Z \rightarrow Z$ – the transformation that executes the task mapping, implemented by a deep network with parameters specified by $\mathcal{H}$. Where language was used to cue meta-mappings, it was processed by language encoder: $\mathcal{L}$ : natural language $\rightarrow Z$.

Each epoch consisted of a separate learning step on each task (both base and meta), in a random order. In each task, the meta-learner would receive only a subset (the "batch size" above) of the examples to generate a function embedding, and would have to generalize to the remainder of the examples in the dataset. The embeddings of the tasks for the meta-learner were computed once per epoch, so as the network learned over the course of the epoch, these embeddings would get "stale," but this did not seem to be too detrimental.

The results reported in the figures in this paper are averages across multiple runs, with different trained and held-out tasks (in the polynomial case) and different network initializations (in all cases), to ensure the robustness of the findings.

### F.3 SOURCE REPOSITORIES

(Will be included in de-anonymized version.)

## G NUMERICAL RESULTS

In this section we provide the mean values and bootstrap confidence intervals corresponding to the major figures in the paper, as well as the baseline results in those figures. Tables were generated with stargazer (Hlavac, 2018).

### G.1 POLYNOMIALS

| named_run_type | is_new | mean_loss | boot_CI_low | boot_CI_high |
|---|---|---|---|---|
| HoMM | Trained | 0.015 | 0.012 | 0.018 |
| HoMM | Held out | 0.246 | 0.188 | 0.308 |
| Untrained HoMM network | Trained | 5.735 | 4.823 | 6.74 |
| Untrained HoMM network | Held out | 5.968 | 4.984 | 6.991 |

Table 2: Table for basic meta-learning, Figure 2a

| named_run_type | result_type | mean_loss | boot_CI_low | boot_CI_high |
|---|---|---|---|---|
| HoMM | Trained mapping, on trained task | 0.094 | 0.091 | 0.098 |
| HoMM | Trained mapping, on held-out task | 1.721 | 1.419 | 2.115 |
| HoMM | Held-out mapping, on trained task | 1.28 | 1.213 | 1.35 |
| HoMM | Held-out mapping, on held-out task | 1.775 | 1.706 | 1.846 |
| Untrained HoMM network | Trained mapping, on trained task | 12.998 | 11.689 | 14.381 |
| Untrained HoMM network | Trained mapping, on held-out task | 15.002 | 13.39 | 16.83 |
| Untrained HoMM network | Held-out mapping, on trained task | 8.36 | 7.898 | 8.833 |
| Untrained HoMM network | Held-out mapping, on held-out task | 8.786 | 8.317 | 9.27 |

Table 3: Table for meta-mapping results from examples, Figure 3a

| named_run_type | result_type | mean_loss | boot_CI_low | boot_CI_high |
|---|---|---|---|---|
| Language | Trained mapping, on trained task | 0.515 | 0.483 | 0.552 |
| Language | Trained mapping, on held-out task | 2.244 | 1.921 | 2.623 |
| Language | Held-out mapping, on trained task | 2.072 | 1.958 | 2.19 |
| Language | Held-out mapping, on held-out task | 2.35 | 2.254 | 2.447 |
| Untrained HoMM network | Trained mapping, on trained task | 13.328 | 11.977 | 14.823 |
| Untrained HoMM network | Trained mapping, on held-out task | 15.313 | 13.602 | 17.354 |
| Untrained HoMM network | Held-out mapping, on trained task | 8.205 | 7.795 | 8.662 |
| Untrained HoMM network | Held-out mapping, on held-out task | 8.625 | 8.131 | 9.104 |

Table 4: Table for meta-mapping results from language, Figure 4a

## G.2 CARDS

| named_run_type | named_game_type | is_new_game | average_reward | avg_rwd_CI_low | avg_rwd_CI_high |
|---|---|---|---|---|---|
| Random 50% holdout | High card | Trained | 0.53 | 0.521 | 0.541 |
| Random 50% holdout | High card | Held out | 0.441 | 0.42 | 0.462 |
| Random 50% holdout | Match | Trained | 0.537 | 0.524 | 0.55 |
| Random 50% holdout | Match | Held out | 0.539 | 0.523 | 0.556 |
| Random 50% holdout | Pairs | Trained | 0.521 | 0.504 | 0.536 |
| Random 50% holdout | Pairs | Held out | 0.453 | 0.434 | 0.47 |
| Random 50% holdout | Straight flush | Trained | 0.525 | 0.508 | 0.54 |
| Random 50% holdout | Straight flush | Held out | 0.484 | 0.466 | 0.502 |
| Random 50% holdout | Blackjack | Trained | 0.582 | 0.557 | 0.603 |
| Random 50% holdout | Blackjack | Held out | 0.492 | 0.468 | 0.513 |
| Targeted 10% holdout | High card | Trained | 0.527 | 0.518 | 0.536 |
| Targeted 10% holdout | Match | Trained | 0.536 | 0.526 | 0.546 |
| Targeted 10% holdout | Pairs | Trained | 0.522 | 0.512 | 0.531 |
| Targeted 10% holdout | Straight flush | Trained | 0.524 | 0.509 | 0.538 |
| Targeted 10% holdout | Straight flush | Held out | 0.361 | 0.332 | 0.39 |
| Targeted 10% holdout | Blackjack | Trained | 0.586 | 0.575 | 0.598 |

Table 5: Table for basic meta-learning, Figure 2b

| is_new_game | named_run_type | named_game_type | expected_reward |
|---|---|---|---|
| Trained | Targeted 10% holdout | High card | 0.531 |
| Trained | Targeted 10% holdout | Match | 0.541 |
| Trained | Targeted 10% holdout | Pairs | 0.532 |
| Trained | Targeted 10% holdout | Straight flush | 0.537 |
| Held out | Targeted 10% holdout | Straight flush | 0.274 |
| Trained | Targeted 10% holdout | Blackjack | 0.592 |
| Trained | Random 50% holdout | High card | 0.531 |
| Held out | Random 50% holdout | High card | 0.396 |
| Trained | Random 50% holdout | Match | 0.541 |
| Held out | Random 50% holdout | Match | 0.541 |
| Trained | Random 50% holdout | Pairs | 0.532 |
| Held out | Random 50% holdout | Pairs | 0.37 |
| Trained | Random 50% holdout | Straight flush | 0.536 |
| Held out | Random 50% holdout | Straight flush | 0.452 |
| Trained | Random 50% holdout | Blackjack | 0.595 |
| Held out | Random 50% holdout | Blackjack | 0.456 |

Table 6: Table for playing most correlated learned strategy for basic meta-learning, dashed colored lines in Figure 2b

| named_game_type | expected_reward |
|---|---|
| Blackjack | 0.592 |
| High card | 0.531 |
| Match | 0.541 |
| Pairs | 0.532 |
| Straight flush | 0.536 |

Table 7: Table for playing optimal rewards for basic meta-learning, solid lines in Figure 2b

| is_new_game | named_run_type | named_game_type | expected_reward |
|---|---|---|---|
| Trained | Targeted 10% holdout | High card | 0.531 |
| Trained | Targeted 10% holdout | Match | 0.541 |
| Trained | Targeted 10% holdout | Pairs | 0.532 |
| Trained | Targeted 10% holdout | Straight flush | 0.537 |
| Held out | Targeted 10% holdout | Straight flush | 0.274 |
| Trained | Targeted 10% holdout | Blackjack | 0.592 |
| Trained | Random 50% holdout | High card | 0.531 |
| Held out | Random 50% holdout | High card | 0.396 |
| Trained | Random 50% holdout | Match | 0.541 |
| Held out | Random 50% holdout | Match | 0.541 |
| Trained | Random 50% holdout | Pairs | 0.532 |
| Held out | Random 50% holdout | Pairs | 0.37 |
| Trained | Random 50% holdout | Straight flush | 0.536 |
| Held out | Random 50% holdout | Straight flush | 0.452 |
| Trained | Random 50% holdout | Blackjack | 0.595 |
| Held out | Random 50% holdout | Blackjack | 0.456 |

Table 8: Table for most correlated baselines for basic meta-learning, dashed colored lines in Figure 2b

| named_run_type | named_meta_task | is_new | average_reward | avg_rwd_CI_low | avg_rwd_CI_high |
|---|---|---|---|---|---|
| Targeted 10% holdout | Switch suits | Trained | 0.523 | 0.512 | 0.534 |
| Targeted 10% holdout | Switch suits | Held out | 0.234 | 0.196 | 0.273 |
| Targeted 10% holdout | Losers | Trained | 0.532 | 0.511 | 0.546 |
| Targeted 10% holdout | Losers | Held out | 0.289 | 0.241 | 0.322 |
| Random 50% holdout | Switch suits | Trained | 0.528 | 0.521 | 0.533 |
| Random 50% holdout | Switch suits | Held out | 0.375 | 0.368 | 0.382 |
| Random 50% holdout | Losers | Trained | 0.531 | 0.523 | 0.538 |
| Random 50% holdout | Losers | Held out | 0.427 | 0.417 | 0.436 |

Table 9: Table for meta-mapping from examples, Figure 3b

| named_run_type | named_meta_task | is_new | average_reward | avg_rwd_CI_low | avg_rwd_CI_high |
|---|---|---|---|---|---|
| Language (random 50%) | Switch suits | Trained | 0.525 | 0.52 | 0.534 |
| Language (random 50%) | Switch suits | Held out | 0.371 | 0.353 | 0.384 |
| Language (random 50%) | Losers | Trained | 0.524 | 0.521 | 0.527 |
| Language (random 50%) | Losers | Held out | 0.426 | 0.413 | 0.44 |
| Language (targeted 10%) | Switch suits | Trained | 0.531 | 0.52 | 0.542 |
| Language (targeted 10%) | Switch suits | Held out | 0.225 | 0.146 | 0.305 |
| Language (targeted 10%) | Losers | Trained | 0.539 | 0.533 | 0.544 |
| Language (targeted 10%) | Losers | Held out | 0.341 | 0.308 | 0.367 |

Table 10: Table for meta-mapping from language, Figure 4b

| named_run_type | named_meta_task | is_new | expected_reward |
|---|---|---|---|
| Targeted 10% holdout | Switch suits | Trained | 0.298 |
| Targeted 10% holdout | Switch suits | Held out | 0.368 |
| Targeted 10% holdout | Losers | Trained | -0.446 |
| Targeted 10% holdout | Losers | Held out | -0.463 |
| Random 50% holdout | Switch suits | Trained | 0.37 |
| Random 50% holdout | Switch suits | Held out | 0.278 |
| Random 50% holdout | Losers | Trained | -0.465 |
| Random 50% holdout | Losers | Held out | -0.444 |

Table 11: Table of rewards if system ignored meta-mapping, colored dashed lines in Figure 3b

