# OpenReview forum: "Zero-shot task adaptation by homoiconic meta-mapping"
_ICLR.cc/2020/Conference — Reject_

### Official Review · AnonReviewer2 · 2019-10-23
**Official Blind Review #2**

**Rating:** 3

**Review:**

In this paper, the authors proposed to address the zero-shot adaptation of tasks by the defined and learned “meta-mappings”. Technically speaking, this work is built upon HyperNetworks, while its major contribution lies in the homoiconic embedding/treatment of a data point, a task, and a meta-mapping. Despite this fascinating and intriguing idea itself, I am still in doubt about its “real” power, considering that a few important references/baselines are missing.

Pros:
-	The ideas of introducing a meta-mapping and treating it similarly or the same as a learning a task itself are novel to me.
-	This work innovatively tackles the problem of zero-shot task adaptation, which is indeed challenging.
-	The case study of the proposed framework in the card game is interesting.

Cons:
-	The major concern for this work is its lack of discussion and comparison with state-of-the-art meta-learning baselines.
    o	In fact, the meta-mapping this work learns is the relationship between tasks, and the meta-task the authors mentioned is a group of tasks. In this sense, I strongly suggest the authors to go through the recent work [1], and compare with it. Even the datasets constructed in that paper can be used as a benchmark to validate the proposed method in image classification.
    o	More recent work on explicitly learning the embedding of a task should be noted and compared, including [1][2]. The meta-mapping in this work is still based on embedding of a task, i.e., z_func, which is obtained by the function network. So is it possible to replace the embedding with recent SOTA algorithms?
    o	Other basic meta-learning algorithms like MAML[3] and many others should still be compared and discussed, especially in the basic meta-learning setting in Figure 2.
-	Many important details and ablation studies are missing, making the work less convincing.
    o	Why don’t merge the results Fig. 2 and Fig. 3 and compare them in a single figure, so that the contribution of meta-mappings can be shown.
    o	How do you train the domain-specific encoder and decoder? Are they generalizable to a wide range of more complicated tasks like image classification?
    o	It would be great to analyze the limitations of the proposed algorithm, especially for the case where a newly coming task belonging to an novel and unseen meta-task arrives. How insightful or predictable is the proposed zero-shot adaptation method in these cases?
    o	How do you obtain the lower-bound (solid line) and the upper-bound (dashed line) of the performance in the figures?
    o	Will the dimension of Z influence the performance? Since data points, tasks, and meta-mappings contain different amount of information, investigating the effect of the dimension of Z is key for the robustness of the method.
-	The paper is too expatiatory with many confusion notations and redundant parts, which make it quite difficult to follow.

References:
[1] Hierarchically Structured Meta-learning, ICML19
[2] TADAM: Task dependent adaptive metric for improved few-shot learning, NeurIPS18
[3] Model-Agnostic Meta-Learning for Fast Adaptation of Deep Networks


**Experience Assessment:**

I have published one or two papers in this area.

**Review Assessment: Checking Correctness Of Derivations And Theory:**

I carefully checked the derivations and theory.

**Review Assessment: Checking Correctness Of Experiments:**

I carefully checked the experiments.

**Review Assessment: Thoroughness In Paper Reading:**

I read the paper thoroughly.

---

> ### Author Response · Authors · 2019-11-11
> **Response to your review (Part 1)**
>
> Thank you for the thorough and thoughtful review. We appreciate that you think our work is tackling zero-shot adaptation in an innovative way! We have made substantial revisions to try to clarify the presentation. We respond to some of your comments inline below.
>
>
> - The major concern for this work is its lack of discussion and comparison with state-of-the-art meta-learning baselines.
>     o   In fact, the meta-mapping this work learns is the relationship between tasks, and the meta-task the authors mentioned is a group of tasks. In this sense, I strongly suggest the authors to go through the recent work [1], and compare with it. Even the datasets constructed in that paper can be used as a benchmark to validate the proposed method in image classification.
> We agree that the approach proposed there is very interesting, and have added a brief discussion of it! However, it does not consider the types of systematic relationships between tasks to which our system appli
> es. For example, switching from winning to losing one of card games results in exactly opposite strategies, while other meta-mappings denote other types of relationships (such as switching what suit is more valuable). Hierarchical clustering cannot capture these richly structure relationships (see e.g. McClelland, Sadeghi, and Saxe (2017), "A Critique of Pure Hierarchy: Uncovering Cross-Cutting Structure in a Natural Dataset"), and hierarchically clustered or similarity based datasets alone thus do not suffice. This is part of why we suggest the construction of new datasets in the future directions.
>
>     o   Other basic meta-learning algorithms like MAML[3] and many others should still be compared and discussed, especially in the basic meta-learning setting in Figure 2.
> Our main contribution is to propose a meta-mapping framework for zero-shot task performance, and parsimonious method for performing these meta-mappings. MAML as such is not a method of zero-shot task performance, it requires examples to learn from. We could therefore compare to MAML for our basic-meta-learning results, but those are simply a sanity check. We also compare to a variety of baselines, including chance and optimal performance, untransformed representations, and the most correlated task experienced (in the cards domain). However, if our paper is accepted, and you feel that the comparison to MAML for basic meta-learning is useful, we will run MAML on our tasks before the camera-ready submission.
>
> One might take inspiration from our framework try to use MAML for zero-shot task performance by transforming task representations would require adopting our meta-mapping framework, as well as a number of ideas of our architecture (where do the task representations come from, and how are they used?), and so its not clear to us that this is an appropriate baseline, rather than simply another implementation of our technique. Instead, we feel the more appropriate baseline is performing tasks from a natural language description, as many prior zero-shot works have, and so we have moved this result to the main text, as noted above.
>
>
> o More recent work on explicitly learning the embedding of a task should be noted and compared, including [1][2]. The meta-mapping in this work is still based on embedding of a task, i.e., z_func, which is obtained by the function network. So is it possible to replace the embedding with recent SOTA algorithms?
>
> Thanks for the useful and interesting references! We believe that it might be possible to replace our method for generating task embeddings with another algorithm. We have added a reference, and we do mention some other methods for constructing function embeddings (last paragraph of the related work).
>
> (continued in next comment, due to character limit)

---

> > ### Author Response · Authors · 2019-11-11
> > **Response to your review (part 2)**
> >
> >
> > - Many important details and ablation studies are missing, making the work less convincing.
> >     We do include a number of ablation studies in Appendix E.
> >
> > `
> > o Why don’t merge the results Fig. 2 and Fig. 3 and compare them in a single figure, so that the contribution of meta-mappings can be shown.
> >     We think this would be extremely misleading, since fig. 2 is meta-learning (viz. performing tasks from examples), while figure 3 is performing tasks zero-shot, without examples. We have tried to edit the captions to clarify this. We have also moved figure 5. to the main text, which does include this comparison between performing a task zero-shot and from examples, as well as another baseline.
> >
> >
> > o How do you train the domain-specific encoder and decoder? Are they generalizable to a wide range of more complicated tasks like image classification?
> >     The model is trained end-to-end. We expect that an appropriate convolutional architecture could be trained end-to-end in our system if given tasks related to image classification, or a pretrained model could be used as an encoder.
> >
> >
> > o It would be great to analyze the limitations of the proposed algorithm, especially for the case where a newly coming task belonging to an novel and unseen meta-task arrives. How insightful or predictable is the proposed zero-shot adaptation method in these cases?
> >     We show in fig. 3 and 4 that the model produces reasonable results even in this case, within the polynomial domain (because we did not have enough meta-mappings in the cards domain to hold one out). Analyzing the features of this generalization in detail seems somewhat beyond the scope of this paper -- indeed, many submissions to this conference are full papers that only perform such analysis on a simpler model.
> >
> >
> > o How do you obtain the lower-bound (solid line) and the upper-bound (dashed line) of the performance in the figures?
> >     We have added some clarification about this to the captions, but briefly:
> >
> >     Polynomial tasks: the dashed chance line is the loss for outputting all zeros, which because of the way our polynomials are sampled is the mean for all polynomials. This is a relatively useful null, e.g. the difference between the dashed line and the model performance can be interpreted as percent variance explained. Optimal performance is zero loss, i.e. regressing every point perfectly.
> >
> >     Cards tasks: the dashed chance line is the loss for choosing actions uniformly at random. The optimal performance is the score achievable by optimal knowledge of the game (that is, betting maximally on every hand that has a positive expected return, and not betting on hands that do not). The colored baselines are various more careful analyses, such as playing the most correlated known strategy, or the performance the model would exhibit if it did not perform its prior task representation.
> >
> >
> > o Will the dimension of Z influence the performance? Since data points, tasks, and meta-mappings contain different amount of information, investigating the effect of the dimension of Z is key for the robustness of the method.
> >     We generally found that too small a Z impaired performance, but once Z was large enough, making it larger generally did not substantially affect performance. This is in line with recent results on overparameterization in deep models.
> >
> >
> > We hope this helps to address your concerns, please let us know if further changes would be useful.

---

### Official Review · AnonReviewer3 · 2019-10-24
**Official Blind Review #3**

**Rating:** 3

**Review:**

This paper presents a method for adapting a model that has been trained to perform one task, so that it can perform a new task (potentially without using any new training data at all—i.e., zero-shot learning). In some ways the presented work is a form of meta-learning or *meta-mapping* as the authors refer to it. The premise of the paper is very interesting and the overall problem is definitely of high interest and high potential impact.

I believe that the presentation of the proposed method can be significantly improved. The method description was a bit confusing and unclear to me. The experimental results presented were all done on small synthetic datasets and it’s hard to evaluate whether the method is practically useful. Furthermore, no comparisons were provided to any baselines/alternative methods. For example, in Sections 4 and 5 I was hoping to see comparisons to methods like MAML. Also, I felt that the proposed approach in Section 5 is very similar to MAML intuitively. This makes a comparison with MAML even more desirable. Without any comparisons it’s hard to tell how difficult the tasks under consideration are and what would amount to good performance on the held-out tasks.

In summary, I feel the paper tackles an interesting problem with an interesting approach, but the content could be organized much better. Also, this work would benefit significantly from a better experimental evaluation. For these reasons I lean towards rejecting this paper for now, but would love to see it refined for a future machine learning conference.

Also, the work by Platanios, et al. on contextual parameter generation is very relevant to this work as it tackles multi-task learning using HyperNetworks. It may be worth adding a short discussion/comparison to that work as it also considers zero-shot learning.

Minor comments:
- Capitalize: “section” -> “Section”, “appendix” -> “Appendix”, “fig.” -> “Figure”. Sometimes these are capitalized, but the use is inconsistent throughout the paper.
- “Hold-out” vs “held-out”. Be consistent and use “held-out” throughout.

**Experience Assessment:**

I have published one or two papers in this area.

**Review Assessment: Checking Correctness Of Derivations And Theory:**

N/A

**Review Assessment: Checking Correctness Of Experiments:**

I assessed the sensibility of the experiments.

**Review Assessment: Thoroughness In Paper Reading:**

N/A

---

> ### Author Response · Authors · 2019-11-11
> **Response to your review**
>
> Thank you for your thoughtful review. We appreciate that you think our problem and approach are interesting! We have made some substantial revisions to clarify the presentation, we hope that these will help address your concerns. We respond to some specific comments below:
>
> > No comparisons were provided to any baselines/alternative methods.
>
> We do compare to a number of lesions in the supplement, and to an alternative method (performing tasks from a description alone). We have moved this latter comparison to the main text at the suggestion of Reviewer 1.
>
> > in Sections 4 and 5 I was hoping to see comparisons to methods like MAML. Also, I felt that the proposed approach in Section 5 is very similar to MAML intuitively. This makes a comparison with MAML even more desirable. Without any comparisons it’s hard to tell how difficult the tasks under consideration are and what would amount to good performance on the held-out tasks.
>
> Our main contribution is to propose a meta-mapping framework for zero-shot task performance, and parsimonious method for performing these meta-mappings. MAML as such is not a method of zero-shot task performance, it requires examples to learn from. We could therefore compare to MAML for our basic-meta-learning results, but those are simply a sanity check. We also compare to a variety of baselines, including chance and optimal performance, untransformed representations, and the most correlated task experienced (in the cards domain). However, if our paper is accepted, and you feel that the comparison to MAML for basic meta-learning is useful, we will run MAML on our tasks before the camera-ready submission.
>
> One might take inspiration from our framework try to use MAML for zero-shot task performance by transforming task representations would require adopting our meta-mapping framework, as well as a number of ideas of our architecture (where do the task representations come from, and how are they used?), and so its not clear to us that this is an appropriate baseline, rather than simply another implementation of our technique. Instead, we feel the more appropriate baseline is performing tasks from a natural language description, as many prior zero-shot works have, and so we have moved this result to the main text, as noted above.
>
> > The experimental results presented were all done on small synthetic datasets and it’s hard to evaluate whether the method is practically useful.
>
> We already have more material than fits in this paper, especially now that we have added clarifications that all reviewers requested. We remained with synthetic tasks for two reasons: 1) to illustrate the method in settings we thought would be clearer, 2) because as we highlight in the future directions, there is a lack of meta-learning datasets that contain as structured of relationships between tasks as we consider. (Taskonomy, for example, has at best a notion of "similarity" in terms of transfer.) We are working on creating several such datasets, but we think that this paper as it stands is a useful contribution that illustrates the concept and ideas -- the datasets themselves will also require further description, and including them in a paper of this length would likely result in even more material being cut, and so a less clear presentation. We do think this is an important direction, but we think this paper makes a useful contribution by highlighting this new perspective. We hope that you agree.
>
>
> > Also, the work by Platanios, et al. on contextual parameter generation is very relevant to this work as it tackles multi-task learning using HyperNetworks. It may be worth adding a short discussion/comparison to that work as it also considers zero-shot learning.
>
> Thanks, we weren't familiar with this very interesting work! We have added a reference and a brief discussion of the relationship.
>
> - Capitalize: “section” -> “Section”, “appendix” -> “Appendix”, “fig.” -> “Figure”. Sometimes these are capitalized, but the use is inconsistent throughout the paper.
>     Thanks for pointing this out, fixed.
> - “Hold-out” vs “held-out”. Be consistent and use “held-out” throughout.
>     We are making a grammatical distinction here -- "held-out" is an adjective and "hold-out" is a noun. That is, one might refer to a "held-out task" or to a "a hold-out." Hopefully this clarifies things.
>
> We hope that this clarifies things, and will help to address your concerns. Please let us know if further changes would be useful.

---

> > ### Comment · AnonReviewer3 · 2019-11-15
> > **Response to Rebuttal**
> >
> > Thank you for your response! You indeed clarified some of my questions so thanks for that. However, I maintain my view that the paper is hard to read and could benefit by some reorganization and restructuring. Furthermore, even though it may be hard to find/create extensive real-world datasets for evaluation, I still believe you can find harder datasets than what is currently considered in the paper. Thus, I'm inclined to maintain my current score. I think refining the paper and resubmitting it would be beneficial for your work and would also help readers appreciate it more.

---

> > > ### Author Response · Authors · 2019-11-15
> > > **Thanks, more detail would be helpful**
> > >
> > > Thank you for replying!
> > >
> > > It would be very useful to have specific suggestions for how to organize the material better, as we have made some changes already at the suggestion of the other reviewers, that were evidently not sufficient for you. What particular aspects of the organization did you think could be improved?
> > >
> > > Also, are there particular datasets that you think would be applicable? We are not aware of any such datasets. We are working on creating some of our own, but again, we feel that our ideas are a useful contribution in part to motivate the creation of such datasets.

---

### Official Review · AnonReviewer1 · 2019-10-26
**Official Blind Review #1**

**Rating:** 3

**Review:**

This paper introduces a zero-shot task adaptation methodology by learning a meta-learner that reuses a latent representation of data points and tasks (Homoiconic Meta-Mapping). The proposed approach is applied to learn multivariate polynomial, a card game and the paper claims that such a method can apply in both supervised and RL settings.

+ves:

+ Task-to-task mapping is a relevant topic in the contemporary context, and the paper seeks to address this relevant problem.
+ The overall idea of considering winning-losing strategies in meta-learning is interesting.
+ The experimental settings of card games and polynomials are interesting to study the proposed setting, and the results are promising.

Concerns:

- A key issue is that the paper claims to be the first to propose a method for zero-shot task adaptation (pg 8, para before “Future Directions”). The paper seems to have missed papers such as (1) Zero-Shot Task Transfer, Pal et. al, CVPR 2019, and, (2) Zero-Shot Task Generalization with Multi-Task Deep Reinforcement Learning, Junhyuk Oh et al, ICML 2017. In particular, the CVPR 2019 paper proposes zero-shot task transfer using meta-mappings. Citing these papers and comparing against them is essential, considering this paper addresses the same objective.

- The proposed methodology does not clearly detail how the zero-shot problem setting is used. Section F.2 (Appendix) gives the formulation of a meta-learner and zero-shot task a bit. However, how the meta-learner performs zero-shot task parameter regression is not clear. The notations used in mathematical equations are also not very clear, and a reader needs to swap between different sections to understand the proposed method. (Moving Sec F.2 to the main paper would be very useful. Similarly, moving Figs 5, 7, 8 etc. to the main paper would have helped greatly too.)

- How is the notion of a “task” defined? Are different classes of a dataset tasks (already done in [3]), or are different vision-related objectives as tasks (reference [1])? Are language-related objectives (such as: document classification, sentiment analysis, Named-entity recognition etc.) tasks? In an RL setting, do zero-shot tasks deal with a new environment? It would have been nice to see clearer definitions for the kind of tasks that this method is suited to.

- It was not clear on how the relationship between source tasks and zero-shot tasks are modeled or computed. Clearly, the performance of such a system depends on the correlations between the tasks, and these details are not clear, especially in generalizing the proposed method. The paper seems to correlate tasks and language-related hints; however getting a language hint is not always available, especially for datasets such as Taskonomy [1, see Refs below] - which is a dataset for multiple tasks.

- The zero-shot part is described using a winning-losing scenario, where we have a partial/side knowledge of winning of task_1 and the model can predict losing strategy. Now, what if:
We do not have any knowledge of the winning strategy (or losing strategy) of a zero-shot task?
Can we predict the losing strategy of a game, let’s say card game, from the knowledge of other games like: soccer, tennis, badminton etc. (the initial motivation of the paper). This is quite possible in the continual learning scenario (the paper mentioned some studies on continual learning). If so, then please comment on the negative transfers in that scenario and how it will affect the inference of the meta-learner.
The proposed method also mentioned language descriptions. In case of tasks that are not easy to describe (say, a vision task such as segmentation), how will the proposed method will compute the task-prior of seen-tasks and zero-shot tasks?
The paper mentions “winning-losing” strategy and the metanetwork gives losing strategy parameters. What if the scenario is the otehr way - given a losing strategy, we now have to perform the winning strategy (IMHO this has more value in the real world). Does the paper show the result on this, or am I missing something?

- Experiments are limited to synthetic data and card games, and it would be nice to see the validation of the proposed method on different domains. The notion of zero-shot tasks are also not clear for multivariate polynomial experiment and card-game experiment.

- Figure 1: please clearly mention in the caption what is the meaning and purpose of probes. It is not completely clear whether probes (also referred as held-out set In figure 1 caption) are taking part in training, or only at the time of validation?

- Shared Z vs. Separate task embedding: reuse of M (or, H network) network will only be possible if the input and output dimensions are same - i.e. the dimension of Z_output of M is the same as the dimension of Z_input of the M. What if  input and output dimensions are not the same?

Other minor comments:

* One general comment is: please mention the “Zero-shot” experiments more clearly (may be a separate paragraph or subsection), otherwise it is hard to follow.
* Figure 2(a) please remove the question mark (?) after Held-out. Similarly, Figure 5: Please complete the sentence “ Performing from …”.
* Figure 1: (a)  Basic meta-learning: Please use some other color to highlight “(a)  Basic meta-learning”, the grey gives an illusion that the step ((a)  Basic meta-learning is not important.
* A term such as “Homoiconic” would have benefited from a formal definition, especially considering it’s not a commonly used term.
* The presentation is in general not well-polished, and needs revision.

References:
[1] Taskonomy, CVPR 2018, Zamir et al
[2] Zero-shot Task Transfer, CVPR 2019, Pal et al
[3] Learning To Detect Unseen Object Classes by Between-Class Attribute Transfer, CVPR 2009, Lampert et al.
[4] Zero-Shot Task Generalization with Multi-Task Deep Reinforcement Learning, ICML 2017, Junhyuk Oh et al.

======= POST-REBUTTAL COMMENTS=========
I thank the authors for the response. Some of the queries/concerns in the review comments above were clarified. I am updating my rating to Weak Reject, but unfortunately am not completely convinced about accepting the paper in its current form for the following reasons:

- Thank you for pointing out the difference between this work and other recent efforts (Oh et al, 2017 and Pal et al, 2019), and updating the manuscript with this content. However, considering these methods have been proposed, it is only natural to ask that the proposed method needs to be compared against them, to judge the usefulness of meta-mappings as against other ways to achieve zero-shot task adaptation, at least on some common datasets/settings. The paper indeed does propose a new idea, but without a comprehensive comparison/evaluation, the inferences on its usefulness may not be conclusive without such studies.

- Considering methods solving the same/similar problem using other approaches have tested on real-world datasets, experiments on synthetic datasets alone seems limiting.  I agree and appreciate the amount of material that is already in the paper, but it is also important to have the correct material to make conclusive inferences that benefit the community.

I once again thank the authors for the efforts in the response, but just think the paper may benefit from revisions based on the comments shared here, to be more impactful/useful.

**Experience Assessment:**

I have published one or two papers in this area.

**Review Assessment: Checking Correctness Of Derivations And Theory:**

I carefully checked the derivations and theory.

**Review Assessment: Checking Correctness Of Experiments:**

I carefully checked the experiments.

**Review Assessment: Thoroughness In Paper Reading:**

I read the paper at least twice and used my best judgement in assessing the paper.

---

> ### Author Response · Authors · 2019-11-11
> **Response to your review (Part 1)**
>
> Thank you for the thorough review. We have made substantial revisions to the paper that we hope will clarify some points and address your concerns. We respond to your comments inline below.
>
>
> - A key issue is that the paper claims to be the first to propose a method for zero-shot task adaptation (pg 8, para before “Future Directions”). The paper seems to have missed papers such as (1) Zero-Shot Task Transfer, Pal et. al, CVPR 2019, and, (2) Zero-Shot Task Generalization with Multi-Task Deep Reinforcement Learning, Junhyuk Oh et al, ICML 2017. In particular, the CVPR 2019 paper proposes zero-shot task transfer using meta-mappings. Citing these papers and comparing against them is essential, considering this paper addresses the same objective.
>
> We did not intend to claim to be the first to perform a task zero-shot, and we have added some review of the papers you cite (and a few others) to the related work section. We do however, maintain that our paper is the first to propose performing a task by transforming a prior task representation from examples of the relationship between the tasks (i.e. the first to propose meta-mapping as such). The paper by Pal et al. does not transform task representations as such, and does not use systematic analogies between tasks, but rather a scalar measure of task similarity (to multiple tasks), which is essentially just another way of describing the task. We believe that functions that take in a task and output another have more representational specificity than scalar similarity, which does not allow for as rich a notion of *along which dimensions* two tasks are similar. We have added some discussion to the paper to clarify this.
>
> - The proposed methodology does not clearly detail how the zero-shot problem setting is used. Section F.2 (Appendix) gives the formulation of a meta-learner and zero-shot task a bit. However, how the meta-learner performs zero-shot task parameter regression is not clear. The notations used in mathematical equations are also not very clear, and a reader needs to swap between different sections to understand the proposed method. (Moving Sec F.2 to the main paper would be very useful. Similarly, moving Figs 5, 7, 8 etc. to the main paper would have helped greatly too.)
>
> We perform a new task zero-shot by transforming the embedding of a prior task according to a meta-mapping. We originally had most of the material from section F.2 in the main text already, but slightly more terse due to the length constraints. We have expanded these sections to hopefully give clearer intuitions, and have tried to clarify a number of aspects of the presentation. We have also included the former Fig. 5 in the main text, which, as you suggest, is an important part of the story. We are not sure we can include further material unless you suggest something we should cut in correspondence.
>
> - How is the notion of a “task” defined? Are different classes of a dataset tasks (already done in [3]), or are different vision-related objectives as tasks (reference [1])? Are language-related objectives (such as: document classification, sentiment analysis, Named-entity recognition etc.) tasks? In an RL setting, do zero-shot tasks deal with a new environment? It would have been nice to see clearer definitions for the kind of tasks that this method is suited to.
>
> A task is a function, i.e. a mapping of inputs to outputs. This is a general definition which can accomodate almost any machine learning framework. For example, the inputs might be images and the outputs classes, or the inputs states, and the outputs Q-values. We therefore think our method may be applicable to many types of tasks, but exploring all applications seems beyond the scope of this paper.
>
> - It was not clear on how the relationship between source tasks and zero-shot tasks are modeled or computed. Clearly, the performance of such a system depends on the correlations between the tasks, and these details are not clear, especially in generalizing the proposed method. The paper seems to correlate tasks and language-related hints; however getting a language hint is not always available, especially for datasets such as Taskonomy [1, see Refs below] - which is a dataset for multiple tasks.
>
> We do not in general rely on language hints. Those were only used as an example. In general, the system can compute a zero-shot embedding of a new task based on examples of the mapping -- i.e. on pairs of input task, output task that implicitly specify the function by which to transform the prior task embedding.

---

> > ### Author Response · Authors · 2019-11-11
> > **Response to your review (part 2)**
> >
> >
> > - The zero-shot part is described using a winning-losing scenario, where we have a partial/side knowledge of winning of task_1 and the model can predict losing strategy. Now, what if:
> > We do not have any knowledge of the winning strategy (or losing strategy) of a zero-shot task?
> > Can we predict the losing strategy of a game, let’s say card game, from the knowledge of other games like: soccer, tennis, badminton etc. (the initial motivation of the paper). This is quite possible in the continual learning scenario (the paper mentioned some studies on continual learning). If so, then please comment on the negative transfers in that scenario and how it will affect the inference of the meta-learner.
> > The proposed method also mentioned language descriptions. In case of tasks that are not easy to describe (say, a vision task such as segmentation), how will the proposed method will compute the task-prior of seen-tasks and zero-shot tasks?
> > The paper mentions “winning-losing” strategy and the metanetwork gives losing strategy parameters. What if the scenario is the otehr way - given a losing strategy, we now have to perform the winning strategy (IMHO this has more value in the real world). Does the paper show the result on this, or am I missing something?
> >
> > The main point of our results is that we are indeed able to predict the losing strategy of a game from knowledge of other games. While generalizing from soccer to card games is beyond the scope of our paper (we only showed mapping of the meaning of "losing" from one card game to another), the hope is that it would be possible. This understanding will not be perfect, however, is that what you mean by negative transfer? In general, even humans cannot perfectly perform a novel task zero-shot. We are simply able to reach reasonable performance, and then learn further from examples.
> >
> > Regarding switching from losing to winning -- in our cards domain, winning and losing are perfectly symmetric, in the sense that if you want to bet on a hand when you're trying to win, you don't want to bet when you're trying to lose, and vice versa. In more complex games, such as chess, this is not true. Winning requires more knowledge than losing, and so we would expect it to be easier to try to lose when you know how to win than to try to win when we know how to lose. We do not think this is a fundamental limitation of our technique, but rather a limitation of the type of knowledge that is available when a task changes. Some things are easier to infer than others. We think exploring how the structure of a task affects its transformability is an important direction for future work.
> >
> > - Experiments are limited to synthetic data and card games, and it would be nice to see the validation of the proposed method on different domains. The notion of zero-shot tasks are also not clear for multivariate polynomial experiment and card-game experiment.
> >
> > We already have more material than fits in this paper. We remained with synthetic tasks for two reasons: 1) to illustrate the method in settings we thought would be clearer, 2) because as we highlight in the future directions, there is a lack of meta-learning datasets that contain as structured of relationships between tasks as we consider. (Taskonomy, for example, has at best a notion of "similarity" in terms of transfer.) We are working on creating several such datasets, but we think that this paper as it stands is a useful contribution that illustrates the concept and ideas -- the datasets themselves will also require further description, and including them in a paper of this length would likely result in even more material being cut, and so a less clear presentation.
> >
> > - Figure 1: please clearly mention in the caption what is the meaning and purpose of probes. It is not completely clear whether probes (also referred as held-out set In figure 1 caption) are taking part in training, or only at the time of validation?
> >
> > As in standard meta-learning, the probe labels are used during training to encourage the system to generalize. We have added a note about this in the caption and clarified further in the text.
> >
> > - Shared Z vs. Separate task embedding: reuse of M (or, H network) network will only be possible if the input and output dimensions are same - i.e. the dimension of Z_output of M is the same as the dimension of Z_input of the M. What if  input and output dimensions are not the same?
> >
> > Because Z is a latent space, we can choose its dimensionality however we wish, so we can choose for those embeddings to be the same. If we have pre-trained encoders we would like to use, we can either choose a latent the same size as their output, or map their latents into Z with an additional layer.

---

> > > ### Author Response · Authors · 2019-11-11
> > > **Response to your review (Part 3, minor comments)**
> > >
> > >
> > > Other minor comments:
> > >
> > > * One general comment is: please mention the “Zero-shot” experiments more clearly (may be a separate paragraph or subsection), otherwise it is hard to follow.
> > >     Thanks, we have noted the sections and clarified this in the figure captions.
> > > * Figure 2(a) please remove the question mark (?) after Held-out. Similarly, Figure 5: Please complete the sentence “ Performing from …”.
> > >     Done.
> > > * Figure 1: (a)  Basic meta-learning: Please use some other color to highlight “(a)  Basic meta-learning”, the grey gives an illusion that the step ((a)  Basic meta-learning is not important.
> > >     We chose this color because our main contribution is about meta-mapping, not meta-learning, so we wanted to highlight the latter panels.
> > > * A term such as “Homoiconic” would have benefited from a formal definition, especially considering it’s not a commonly used term.
> > >     There are debates about how to precisely define this term (see e.g. https://www.expressionsofchange.org/dont-say-homoiconic/), so we didn't want to wade into such a definitional conundrum. However, we provide a statement of the general agreed-upon part of the definition in the introduction.
> > > * The presentation is in general not well-polished, and needs revision.
> > >     Are there specific points that would benefit from further clarification, beyond what you have mentioned above?

---

### Decision · Program_Chairs · 2019-12-19

**Decision:**

Reject

**Comment:**

The authors presents a method for adapting models to new tasks in a zero shot manner using learned meta-mappings.  The reviewers largely agreed that this is an interesting and creative research direction.  However, there was also agreement that the writing was unclear in many sections, that the appropriate metalearning baselines were not compared to, and that the power of the method was unclear due to overly simplistic domains.  While the baseline issue was mostly cleared up in rebuttal and discussion, the other issues remain.  Thus, I recommend rejection at this time.